# The Role of microRNAs in the Mammary Gland Development, Health, and Function of Cattle, Goats, and Sheep

**DOI:** 10.3390/ncrna7040078

**Published:** 2021-12-13

**Authors:** Artem P. Dysin, Olga Y. Barkova, Marina V. Pozovnikova

**Affiliations:** Russian Research Institute of Farm Animal Genetics and Breeding–Branch of the L. K. Ernst Federal Science Center for Animal Husbandry, Pushkin, 196625 St. Petersburg, Russia; barkoffws@list.ru (O.Y.B.); pozovnikova@gmail.com (M.V.P.)

**Keywords:** miRNA, mammary gland, farm animals, lactation

## Abstract

Milk is an integral and therefore complex structural element of mammalian nutrition. Therefore, it is simple to conclude that lactation, the process of producing milk, is as complex as the mammary gland, the organ responsible for this biochemical activity. Nutrition, genetics, epigenetics, disease pathogens, climatic conditions, and other environmental variables all impact breast productivity. In the last decade, the number of studies devoted to epigenetics has increased dramatically. Reports are increasingly describing the direct participation of microRNAs (miRNAs), small noncoding RNAs that regulate gene expression post-transcriptionally, in the regulation of mammary gland development and function. This paper presents a summary of the current state of knowledge about the roles of miRNAs in mammary gland development, health, and functions, particularly during lactation. The significance of miRNAs in signaling pathways, cellular proliferation, and the lipid metabolism in agricultural ruminants, which are crucial in light of their role in the nutrition of humans as consumers of dairy products, is discussed.

## 1. Introduction

Milk is one of the most important biological fluids for the lives of mammalians over the course of evolution, assuming the role of the first source of energy and nutrients in the life cycles of living organisms, necessary to ensure their proper growth and development. Among the substances contained in milk are both macronutrients (proteins, lactose, and lipids) and micronutrients (vitamins and minerals), between which a certain balance is observed, as well as a number of other bioactive components that are of particular benefit to the body in the early stages of life. Therefore, milk contains growth factors such as epidermal growth factor, neuronal growth factors, VEGF (vascular endothelial growth factor), erythropoietin, growth-regulating factors, and immune-related factors (immune cells, cytokines, chemokines, etc.) [1]. This complex of components is produced as a result of such a composite and dynamic process as lactation, which takes place in the mammary gland. Many variables impact the lactation process, including genetic, epigenetic, non-genetic, and environmental influences. The control of lactation is not only essential for increasing milk production and quality, but it also serves as a model for fundamental cellular processes (proliferation, differentiation, survival, and apoptosis) [2] that may have consequences for productivity (milk yield) and disease state (e.g., mastitis and breast cancer). Over time, more and more data on the endocrine regulation and signaling pathways underlying the physiological processes that occur in the breast are becoming available [3,4,5,6,7].

Lactation is accompanied by changes in the gene activity in the mammary gland. The creation of an assembly of the *Bos Taurus* genome, along with studies of the proteome and gene expression, has made it possible to estimate the number of genes involved in milk production, from mammogenesis to milk secretion [8,9]. Between 6000 and 19,000 genes distributed across all 29 autosomes and the X chromosome of cattle have been reported to be differently expressed during the lactation cycle [10,11]. Numerous genome-wide association studies (GWASs) using high-density SNP chip data have previously been conducted to narrow down areas and identify causal genes that affect milk productivity traits [12,13,14,15,16,17,18].

In recent decades, the regulation of mammalian genes has become a much more compelling issue for researchers than the central dogma of molecular biology. Less than 2% of the mammalian genome contains protein-coding regions, and much larger noncoding RNAs (ncRNAs) are transcribed [19]. Although ncRNAs are grouped into several classes based on transcript size, more data indicate that this group of RNAs is extensive and varies in a similar manner to their protein-coding mRNA analogues [20].

miRNAs are an extensive class of short ncRNAs with a length of approximately 22 bp, first detected in *Caenorhabditis elegans* in 1993 [21]. They regulate multiple cellular processes through the post-transcriptional repression of gene expression, by binding to the 3’-UTRs of mRNAs and inhibiting translation initiation or elongation, as well as inducing co-translational protein degradation [22,23]. This can be realized by mechanisms such as cap inhibition, the inhibition of 60S subunit attachment, ribosome precipitation, P-body sequestration, mRNA decomposition, and mRNA stabilization, inhibiting translation [24]. Since the partial complementarity of miRNAs is sufficient to target mRNAs, each miRNA is able to regulate a large set of genes [25]. It is also known that most miRNA genes located in the introns of protein-coding genes share a common host gene promoter [26]. At the same time, there is growing evidence that the same miRNA can bind to multiple regions and affect the expression of different genes [26,27]. Thus, according to various data, miRNAs control the activity of between 30% [28] and 60% [23] of all protein-coding genes and are involved in the regulation of almost all investigated mammalian cellular processes, such as development, immune system activity, proliferation, neoplastic transformation, and apoptosis [29].

Since the discovery of the first miRNA, lin-4, thousands of miRNAs have been discovered in humans, mice, farm animals, and plants, thanks to deep-sequencing technology and developments in the bioinformatic processing of deep-sequencing data. In light of the crucial regulatory role of miRNAs in many biological processes in different species, they are considered candidates for biomarkers of various human diseases, such as autoimmune [30], metabolic [30], and cardiovascular diseases [31], as well as various types of cancer [32]. Knowledge of the participation of miRNAs in most biological processes in a living organism, as well as the uniqueness of how the mammary gland evolves through cycles of cellular proliferation, differentiation, and apoptosis, provides an opportunity to generalize and adequately characterize the roles of these epigenetic elements in the processes that occur in the ruminant mammary gland in different physiological states, which may provide new insights into the post-transcriptional regulation of the gene expression of this organ.

In this review, an overview of the latest studied aspects of miRNA, directly related to mammary gland function and, in particular, to the process of lactogenesis, is provided. No less attention is paid to the functions of miRNAs in the mammary gland in its development.

## 2. Locations of miRNAs in the Mammary Glands of Various Animals

The paramount question in assessing the role of miRNAs in the lactation process is whether the miRNAs present in milk originate from the blood or are mammary gland-specific. To address this issue, Chen et al. (2010) compared the miRNA profile in milk to that of serum from healthy cows, and found that the total amount of miRNAs in the milk was approximately twice as high as that in the serum; they also identified 47 miRNAs unique to milk [33]. Human breast milk also has a different pattern of miRNA expression compared to blood plasma [34]. These results clearly show that alveolar mammary gland cells express their own miRNAs.

The presence of miRNAs in the mammary gland cells of mice [35], humans [36], and cows and other farm animals [37,38,39] at different stages of development is now known, suggesting that they play an important role in the differential expression involved in mammary gland development and lactation. In addition, in recent years, with the development and expansion of the use of RNA sequencing technology for profiling the expression of miRNAs of different domestic mammals [40], more and more data on the role of miRNAs in various livestock species [41,42,43], the main source of human milk consumed en masse, have begun emerging. Thus, the most recent miRBase release contains 1025 mature miRNAs for cattle, 436 for goats, and 153 for sheep [44], while the RumimiR database, according to the latest data, contains 6808, 4011, and 9276 miRNAs, respectively [44]. In this regard, it is advisable to first consider the various manifestations of miRNAs in the lactation mechanisms in some types of farm animals.

### 2.1. Cattle

Using various high-tech approaches, such as nucleotide microarrays [45], genome sequencing, and RNA sequencing [46], it has been possible to obtain and study miRNA profiles in the mammary tissue and milk of cows. After the full genome sequencing of cattle, 496 miRNA genes were identified, 135 of which were new [46]. The expression profiles of the miRNAs in mammary tissues and cells facilitate the discovery of new miRNAs, as well as the identification of miRNA candidates for different cell types and lactation stages, periods, disease responses, etc. Shortly before obtaining the complete bovine genome sequence, Gu et al. (2007) first discovered miRNAs in the mammary glands of cattle by cloning and sequencing small RNAs from the mammary gland tissue, followed by the identification of 59 individual bovine miRNAs [47]. A total of 31 miRNAs from other bovine tissues were identified through homology searches, 13 of which turned out to be new [48]. Using next-generation sequencing methods, Chen et al. (2010) identified 230 and 213 known miRNAs in the colostrum and mature milk of cows, respectively [33]. The authors also observed that, in the colostrum, compared to in the mature milk, 108 miRNAs were activated and eight were suppressed [33].

Using a microarray, Izumi et al. (2012) identified 100 and 53 known miRNAs in the colostrum and mature milk, respectively [49]. Moreover, using Solexa sequencing, Li et al. (2012) reported 884 unique miRNA sequences in the mammary gland of cattle (283 known, 505 new, and 96 conserved miRNAs). A total of 56 miRNAs in lactating mammary glands showed significant differences in expression compared to in non-lactating mammary glands [50]. Le Guillou et al. (2014) identified 167 new miRNAs in the mammary glands of cows, many of which were also found in the mammary glands of mice [51]. In mammary epithelial cells infected with *Escherichia coli* or *Staphylococcus aureus*, Jin et al. (2014) were able to detect 231 known and 113 new miRNAs, the expression of which was regulated by pathogenic bacteria [51]. Via the analysis of 361 million sequence reads, Li et al. (2015) obtained 321 already-known and 176 new miRNAs [52]. Analyzing three milk fractions (fat, serum, and cells) and breast tissue, Li et al. (2016) reported 210, 200, and 249 known and 33, 31, and 36 new miRNAs in milk fat, serum, and cells, respectively, as well as 321 known and 176 new miRNAs in breast tissue [53].

Wicik et al. (2016) found 54 miRNAs actively expressed in the mammary cells of both dairy and beef breeds [54]. Meanwhile, 292 known and 116 new miRNAs were detected in the mammary epithelial cells of Chinese Holstein breeds by Solexa sequencing [55]. After conducting the deep sequencing of milk fat along the lactation curve, they also identified a total of 475 known and 238 new miRNAs [56]. Illumina/Solexa high-throughput sequencing technology was used to identify approximately 259 miR families, 359 mature miRNAs, 363 pre-miRNAs, 230 new miRNAs, and five specific miRNAs that were expressed in Chinese water buffalo mammary gland tissues [57]. Ju et al. (2018) found 383 loci corresponding to 277 known and 49 putative new miRNAs, two potential mitrons, and 266 differentially expressed miRNAs in the mammary glands of healthy and mastitic cows [58]. miRNAs related to bacterial bovine mastitis were first identified by Jin et al. in mammary cells infected with heat-inactivated *S. aureus* and *E. coli.* A total of 231 known and 113 new miRNAs were identified in this study, including miR-21-5p, miR-27b, miR-22-3p, miR-184, let-7f, miR-2339, miR-499, miR-23a, and miR-99b, which were specific for cells infected with *S. aureus* [59]. Luoreng et al. (2018) sequenced 1838 miRNAs from mastitis-affected breast tissue samples, including 580 known miRNAs (included in the miRbase database) and 1258 predicted new miRNAs [60]. In addition, 492 known and 980 new miRNAs were identified in the milk exosomes of *S. aureus*-infected cows, of which 22 and 15, respectively, showed differential expression compared to those of healthy cows [61]. Han et al. (2020) identified 48 miRNAs in the udders of mastitic cows, 10 of which were known [62].

Using mRNA sequencing in the mammary glands of lactating dairy cows, Cui et al. (2020) identified 497 known miRNAs and 49 new ones. Among these, miRNA-71 was expressed differently in cows with high and low protein and fat contents in their milk [63]. Combined with their previous RNA sequencing data, 21 differentially expressed genes can be called targets for some of the 71 differentially expressed miRNAs, indicating that they may play a critical role in regulating milk protein and fat properties in dairy cattle [64].

To clarify the above information on the discovered bovine miRNAs, we have summarized it in Table 1

### 2.2. Goats

In total, 436 miRNAs have been identified during the sequencing of the goat genome according to miRBase data [65], or 4011 according to RumimiR. However, the most miRNAs have been found on chromosome 21 [66].

Ji et al. (2012) discovered 131 novel and 300 conserved miRNAs in goat mammary tissue during early lactation using Illumina/Solexa high-throughput sequencing [38]. Li et al. (2012) found 346 known and 95 novel miRNAs in goat mammary tissue following desiccation and peak lactation using the same approach (Illumina/Solexa sequencing) [67], while Hou et al. (2017) identified 57 known and 74 potential new miRNAs in colostrum and at peak lactation using high-throughput sequencing followed by analysis [68]. At the same time, using Solexa sequencing with bioinformatics analysis, the same authors were able to identify 1113 conserved known miRNAs and 31 potential new miRNA candidates [69]. However, later, by conducting a study of breast tissue at the same stages of lactation, as well as at the stage of pregnancy, Xuan et al. (2020) found 4038 different miRNAs, which is even more than in the RumimiR database [70]. In another study, where the de novo approach was used to identify miRNAs, by comparing the obtained sequences with already-known goat sequences, 924 miRNAs were detected, while 1178 were found in comparison to the sequences obtained from cattle [71]. Using bioinformatics approaches, such as expressed sequence tag analysis and genome sequence analysis, 29 mature miRNAs were identified and verified [72]. Ji et al. (2017) managed to detect 1487 miRNAs unique to goats, 45 of which had not been identified before [73].

Goat miRNA data are also supplied in an individual table (Table 2).

### 2.3. Sheep

Most of the miRNAs identified in sheep come from tissues other than the mammary gland. For example, Caiment et al. (2010) [74] identified 747 miRNAs from skeletal muscle through deep sequencing, while McBride et al. (2012) [75] reported 212 miRNAs from the ovarian follicles and corpus luteum of sheep at different reproductive stages. Galio et al. (2013) showed the presence of three known miRNAs, including miR-21, miR-205, and the miR-200 family, and 47 novel miRNAs in pregnant and lactating sheep [76]. Recently, Wang et al. (2021) found 147 miRNAs in sheep mammary tissue, studying their expression depending on the period [39].

The sheep miRNA data are also represented in the table.

The above analysis of the chronologically presented studies dedicated to the presence of miRNAs in the mammary glands of the most commonly used farm species and the attempts to compile their complete profiles and miRNAomes shows the different effectiveness of various approaches to miRNA identification. This should help in selecting the most promising method that will generate results most closely representing the truth and providing as complete an miRNA landscape as possible. If the number of miRNAs identified by an approach is taken as a measure of that approach’s efficacy, we must conclude that modern methods, such as Solexa sequencing, as well as the analysis of comparisons of miRNAs in the same tissues but in other species, remain the most promising today. However, the most noteworthy aspect in our framework is that it was feasible to establish the presence of a large number of miRNAs in the tissues of the mammary glands of the most widely used types of dairy livestock using all of the methods discussed above. This should help to facilitate better knowledge of how the mammary gland works, highlighting the new aspects that underpin it. As a result, the particular findings of determining the role of several of the identified miRNAs will be addressed further in the scope of this work (Table 3).

## 3. Regulation of miRNA Gene Expression in the Mammary Glands of Ruminants

The mammary gland is an important organ for studying gene expression since it is the organ primarily responsible for the lactation process. As in many organs, during different periods of life, the gene expression in its cells also differs, as the mammary gland undergoes different cycles of differentiation and regression in adulthood. In general, in any eukaryotic organism, the regulation of gene expression is a complex process that includes DNA methylation, chromatin modification, imprinting, and interfering RNA [77]. As previously emphasized, miRNAs regulate gene expression at the post-transcriptional stage, either by causing the degradation of RNA or by blocking translation by pairing complementarily with bases within mRNAs. Since the partial complementarity of miRNAs is sufficient to target mRNAs for repression, each miRNA has the ability to regulate a large number of genes [78]. Another epigenetic feature is that conserved miRNAs can regulate different genetic pathways and developmental processes in different organisms [27]. In this regard, it is important to fully characterize, in this section, the mechanism of the function of miRNAs in mammary cells under different physiological conditions, which may provide a new understanding of the regulation of the expression of the genes responsible for the lactation process.

### 3.1. Functions of Mammary miRNAs and the Variability of Their Expression Depending on the Reproductive Period

The second most important issue after the local specificity of the miRNAs involved in lactation for mammary gland tissue is the question of the consistency of the miRNA expression patterns in the mammary gland throughout the lactation period. At different stages of the development of the mammary gland, its structure and physiological function change accordingly, and these changes are affected by different hormones, genes, and regulatory factors. As important regulatory factors for gene expression, miRNAs play a critical role in many aspects, including metabolism, the onset and course of disease, mammary gland development, and lactation regulation. As already noted in the previous section, during different periods of lactation, researchers have detected different miRNA expression profiles, which should indicate that the expression of miRNAs is variable. One of the first studies of the differences in miRNA expression depending on mammary gland development was that by Avril-Sassen et al. (2009), who found 102 miRNAs in the mammary gland of mice and observed differences in the time of their expression, on the basis of which they were divided into seven groups [35].

However, of greater interest to us are the further studies carried out on dairy farm animals. Thus, of the 1,692,810 reads obtained by Li et al. (2012) for cattle, 34% corresponded to miRNAs expressed only during the drying period compared to the peak period of milk production. Moreover, analysis of the expression patterns of 173 differentially expressed miRNAs has shown that 165 were suppressed during peak lactation compared to dry periods. Among the sequences described by Li et al. (2012), 56 showed significant differences in expression between lactating and non-lactating cows, as assessed using the IDEG6 package [50]. Nine of these were expressed only in lactating animals, and six in non-lactating animals. However, 48 of these have been confirmed by deep sequencing [50], indicating that deep sequencing may be more sensitive and reliable than microarray analysis in identifying differentially expressed miRNAs. Thus, the miR profiling of lactating and non-lactating bovine mammary glands conducted by Li et al. (2012) revealed decreased expression of miR-125b, -181a, and -199b during the non-lactation period, whereas the expression of miR-141, -484, and -500 was higher during lactation [50]. Similarly, Wang et al. identified 12 downregulated miRNAs in the dry period (30 days before delivery) compared to the early lactation period (seven days before delivery) and one activated miRNA at the beginning of lactation compared to the dry period [45]. Later, Do et al. investigated the miRNA expression pattern during the lactation cycle to study its regulatory mechanisms during lactation using milk fat as an input tissue for sampling. To do this, they examined samples taken at lactogenesis (days 1 and 7), galactopoiesis (days 30, 70, 130, 170, and 230), and involution (day 290, or when the milk yield was reduced to 5 kg/day) in nine cows for deep sequencing. They observed that 15 miRNAs were highly expressed in all the lactation stages; miR-148a and miR-26a were the most widely expressed, accounting for more than 10% of the reads at each stage of lactation. The authors also performed differential expression analyses and found that miR-29b/miR-363 and miR-874/miR-6254 were important mediators of transient signals from lactogenesis to galactopoiesis and from galactopoiesis to the involution stage, respectively. In addition, differential expression analysis showed different patterns of miRNA expression along the lactation curve. For example, some miRNAs were highly expressed during early lactation (lactogenesis) with a subsequent decrease in expression in later stages, whereas others were weakly expressed during early lactation, but showed increased expression in mid-lactation and decreased expression during late lactation, and vice versa [79]. When comparing post-pubertal beef heifers (limousine) to dairy cattle (Holstein–Friesian breed), the level of miR-2285 was substantially higher in the mammary gland tissue of Holstein–Friesians [54]. However, its function in the mammary gland remains unknown.

The temporal pattern of miRNA expression has been reported in other ruminant species. For example, Galio et al. reported changes in the expression pattern of miR-21, miR-205, and the miR-200 family in the mammary tissues of pregnant and lactating sheep. The expression of miR-21 and miR-25 was decreased in early, intermediate, and late pregnancy and during lactation, whereas the miR-200 family (miR-200a, miR-200b, miR-200c, miR-141, and miR-429) showed increased expression [76]. Li et al. found 15 differentially expressed miRNAs in goats when comparing peak lactation and dry periods, including three highly expressed miRNAs (miR-451, miR-2478, and miR-2887) during peak lactation and 12 highly expressed miRNAs (miR-25, miR-128, miR-93, miR-98, miR-145, miR-199b, miR-199a-3p, miR-181b, miR-222, miR-221, let-7b, and let-7c) during the dry period, when examining the miRNA expression patterns during early and peak lactation and in the dry periods [67]. However, in cattle, miR-221 is highly expressed in the early lactation period as compared to the novelty and dry periods [80]. One of the most grandiose studies of goat mammary miRNA was carried out by Xuan et al. (2020), mentioned in the previous chapter. In addition to identifying more than 4000 miRNAs expressed during mammary gland development, they also established the periods of their expression—only 2988 miRNAs were expressed during all three stages of development, and the expression levels of miRNAs such as miR-148a-3p and miR-30d in the dry period were significantly lower than those in late lactation and pregnancy, while the expression levels in the late lactation stage were the highest [70]. This contrasts with the observations of Chen et al. (2017), according to whom miR-148a, like miR-17-5p, is highly expressed in the mammary glands of goats during the early and dry periods [81]. As also already mentioned, Li et al. (2012) reported sequences of highly productive members of the let-7 family found during the dry period and at peak lactation [67], and Hou et al. (2017), in addition to identifying new miRNAs, observed that 45 miRNAs were significantly involved in colostrum during breast feeding, while 86 were almost completely suppressed compared to peak lactation [68]. Chen et al. (2016) also found that the expression of 54 miRNAs changed between peak and late lactation [82]. Chu et al. found that miR-15b was expressed differently at different stages of mammary gland development in mice and goats, showing reduced expression during lactation [83]. The results of Chen et al. (2018) showed that miR-135b, which suppresses the prolactin hormone, is highly expressed in the mammary glands of goats during early and late lactation [84]. Lin et al. (2013) found that the expression of miRNA-27a was 1.25 times higher in the middle of lactation than in the dry period [85]. Taken together, these data indicate that the nature of miRNA expression varies depending on the physiological state of the animal during different reproductive periods.

However, we are also interested in the roles of these miRNAs in the body and the reasons they are expressed in particular periods. Thus, to study the expression of specific miRNAs associated with cell proliferation, metabolism, and innate immune responses during lactation, Wang et al. (2012) evaluated the expression of 13 miRNAs in cows during the dry period (30 days before calving), during the newly calved period (seven days after birth), and at the beginning of lactation (30 days after labor). Twelve identified miRNAs (miR-10a, -15b, -16, -21, -33b, -145, -146b, -155, -181a, -205, -221, and -223) were suppressed during the dry period compared to during lactation. The exception was miR-31, which showed greater expression in the early lactation period compared to the dry period [67]. It has been proposed that miR-146b can modulate Sirtuin1, inhibiting negative adipogenesis regulators and thus boosting adipogenesis [86]. The upregulation of miR-146b during pregnancy was observed, especially in lumen precursors compared to basal/stem cells, indicating that it is involved in the differentiation of mammary epithelial cells. Moreover, the expression of miR-146b is enhanced in lumen precursors in pregnant mice, indicating that miR-146b is involved in the differentiation of mammary stem cells [87].

Under normal conditions and using bioinformatics analyses and biological experiments, Xue et al. (2013) demonstrated that miR-31 activated IL-2 (interleukin 2) expression by decreasing the levels of cytokines upstream of the kinase suppressor KSR2 (ras 2 kinase suppressor) [88]. Interleukins are present in breast milk [89] and play an important role in modulating the immunological systems of offspring [90]. These data indicate that 31 miRNAs may play indirect immunological roles in newborns.

Galio et al. (2012) discovered three primary temporal patterns of miRNA expression in sheep using microarray analysis [76]. Pattern 1 expression was reduced during pregnancy, pattern 2 miRNA expression was induced during pregnancy, and pattern 3 miRNA expression was induced during breastfeeding, according to the findings. The authors took one miRNA from each sample and used RT-qPCR to validate its expression in four animals. In non-pregnant sheep and sheep early in pregnancy, miR-21, which is expressed in alveolar epithelial cells, was active. The involvement of miR-21 in adipogenic differentiation has been linked to this expression pattern. In this context, Kim et al. (2009) discovered that TGF signaling regulated miRNA-21 activity in adipogenic tissue. On the contrary, miR-205 is primarily expressed in the basal membranes of normal mammary gland ducts and lobules during the first half of pregnancy, whereas miR-200 is produced in epithelial cells throughout pregnancy but is activated at the end of pregnancy during breastfeeding [91]. Recently, Wang et al. (2021) conducted the first study reporting systematic miRNA expression profiles in sheep mammary tissue, obtaining totals of 19.9 and 20.1 million net reads during peak lactation and in the no-lactation period, respectively, and finding 147 differentially expressed miRNAs at the peak of lactation and during the dry period. miR-148a, miR-30a-5p, and miR-103 showed higher levels of expression at the peak of lactation than in the non-lactation period, while the expression of miR-143, miR-21, miR-26a, miR-148a, and let-7f dominated in the non-lactation period [39].

Having shed light on the differences in the expression of distinct mammary miRNAs, we can conclude that this issue is crucial to consider when researching the development and function of this organ. The key theses of this chapter are reprinted in Table 4.

### 3.2. Hormonal Regulation of Mammary miRNAs

Mammary gland epithelial cells comprise a single layer of cells that surround the lumen of the alveolar structures and produce milk [92,93]. miRNA expression in mammary gland epithelial cells is regulated by lactogenic hormones (dexamethasone, insulin, and prolactin), as evidenced, for example, by an increased level of miR-148a in these cells in cattle, which is probably associated with increased milk production during lactation in cows [94]. Therefore, to better understand the temporal patterns of miRNA expression, it is important to study miRNAs’ interactions with major hormones. Prolactin, a key hormone that regulates lactation, has been found to promote the expression of miR-23a, miR-27a [85], miR-27b, miR-103, and miR-200a [76,95]. At the same time, prolactin can inhibit the expression of miR-183 [96], and Chen et al. (2018) found that miR-135b, in turn, could suppress prolactin itself and studied the mechanism of its inhibition in detail [84]. In mice, miR-138 has been found to regulate mammary development and galactopoiesis by acting on the prolactin receptor, thus modulating the physiological role of prolactin in mammary cells. miR-135a was also able to suppress the prolactin receptor gene in a culture of goat mammary epithelial cells [97]. Muroya et al. (2016), in turn, through the treatment of cultures of epithelial cells of the mammary glands of cattle with a complex of lactogenic hormones, observed that the expression of miR-21-5p, miR-26a, miR-320a, and miR-148a in milk was lower in the hormone-treated cells than in untreated cells, whereas, by contrast, the expression of miR-339a was lower in the hormone-treated cell-culture medium, which also indicates a different intensity of expression of miRNAs during different periods of mammary development [94]. The steroid hormones estradiol and progesterone also reduce miR-15b expression, with a subsequent increase in lipid formation in mammary epithelial cells [83]. At the same time, miR-126-3p inhibits the expression of the progesterone protein in the epithelial cells of the mammary glands of mice [98].

Growth hormone is the most important galactopoietic hormone in ruminants [99] and triggers casein expression [100]. The results of the above studies show that miR-15a indirectly reduces milk production by blocking the expression of the growth hormone receptor, thus highlighting a new mechanism of growth hormone receptor regulation [101]. The TGF-β/miR-424/503 axis is part of the mechanism that regulates the proliferation of hormone receptor-positive mammary epithelial cells in vivo [102]. Bioinformatic analysis using the RNAhybrid software identified the target miR-15a sequence on the growth hormone receptor’s mRNA [101]. However, Chen et al. (2010) failed to identify this miRNA when searching for the sequence [33]. To confirm this conclusion, Li et al. (2012) transfected this small RNA into mammary epithelial cells and, as a result, observed a decrease in growth hormone receptor transcription and β-casein expression [101].

Thus, a thorough study of the functions of miRNAs and the mechanisms of their action contributes to a better understanding of the physiological and endocrinological conditioning of the processes occurring during lactation. Table 5 summarizes the essential details of the interactions between miRNAs and hormones.

### 3.3. Lipid Metabolism

Lipids are the main component of milk, as evidenced by the synthesis and secretion of more than 30 g of triglycerides, 12 g of proteins, and 5 g of lactose in the mouse mammary gland during the 20-day lactation period, with lipids providing 50% of the energy needs of the offspring organism [92,103,104]. In addition, the adipose tissue in mammals is the main source of energy for the synthesis of milk fat [105]. Epigenetic mechanisms, including DNA methylation, histone modifications, and the regulation of ncRNAs, are involved in the regulation of lipogenesis [106]. In particular, it has been shown that the histone deacetylases and sirtuins, including SIRT1–7, also play a central role in lipid metabolism [107,108,109]. Most studies have used dual luciferase reporter analysis to establish the targeting relationship between miRNAs and their target genes, after the transfection of miRNAs and vectors bearing the 3′-UTRs of interest into breast epithelial cell cultures.

Lin et al. (2013) found the first evidence for the miRNA-mediated regulation of milk fat by miR-103, which can also reduce β oxidation by regulating the AMP-activated protein kinase α (AMPKα) pathway, thus promoting triglyceride accumulation and thereby controlling the ratio of unsaturated/saturated fatty acids in goat’s milk [110]. Recent studies have confirmed that miRNAs also play an important role in lipogenesis and triglyceride homeostasis [111]; triglycerides make up at least 98% of milk fat [112]. The role of miRNAs as regulators of lipogenesis has been confirmed by overexpression studies, which have shown greater synthesis of fat droplets, the accumulation of triacylglycerols, and a higher proportion of unsaturated fatty acids in lactating mammary epithelial cells. For example, miR-24 has been found to be expressed at a much higher level during peak lactation in goats and affects the triacylglycerol content, the unsaturated fatty acid concentration, and the expression of target genes such as FASN (fatty acid synthase), SREBF1 (sterol regulatory element binding transcription factor 1), SCD (stearoyl-CoA desaturase), GPAM (glycerol-3-phosphate acyltransferase; mitochondrial), and ACACA (acetyl-CoA carboxylase) [113]. Moreover, increased expression of miR-200a suppresses the mRNA expression of the genes involved in fat-droplet formation [95]. It also cannot be overlooked that one member of the miR-212 family, miR-212-5p, specifically binds to the 3’-UTRs of stearoyl-CoA of desaturase-1 (SCD1) and fatty acid synthase (FAS) and inhibits their activity, while the overexpression of miR-212-5p reduces the levels of the proteins SCD1 and FAS in vitro and in vivo, while the suppression of miR-212-5p has the opposite effects in primary mouse hepatocytes [114]. miR-135b is functionally related to triacylglyceride synthesis through modulating the large tumor suppressor gene 2 (LATS2) [84]. Another upstream regulator of LATS2 is miR-497, which can thus inhibit the production of triglycerides and unsaturated fatty acids in the epithelial cells of the mammary glands of cattle [115]. Introducing a miR-16a mimic into bovine mammary epithelial cells, according to Chen et al. (2019), in turn, disrupts fat metabolism by targeting the large tumor suppressor kinase 1 (LATS1) [116].

miR-375 also promotes lipogenesis in mouse preadipocytes by regulating ERK1/2 signaling upstream of the peroxisome proliferator gamma receptor (PPARγ) [117], a major regulator of fat-cell formation and differentiation, and is involved in lipid metabolism [118], while miR-146b can promote lipogenesis by suppressing SIRT1, which interferes with the SIRT1–FOXO1 cascade [86]. miR-204-5p and miR-141 also promote lipid synthesis in mammary epithelial cells by modulating SIRT1, as well as SREBF1, FASN, and PPARγ [119,120]. In their study, Lu et al. (2020) found a pro-adipogenic miRNA, miR-212, which can significantly promote lipogenesis by suppressing SIRT2 and regulating the expression of FASN and sterol regulatory element binding protein-1 (SREBP1), thereby increasing the fat content in bovine mammary epithelial cell lines [121]. miR-106b can bind the 3’-UTR of ATP binding cassette subfamily A member 1 (ABCA1), a gene previously identified as positively associated with the synthesis of cow’s milk fat [122], and can reduce the accumulation of triglycerides and cholesterol in the epithelial cells of the mammary glands of large cattle [123]. Chen et al. (2016) also demonstrated that the overexpression of miR-30e-5p and miR-15a in goat mammary epithelial cells promoted fat metabolism via LRP6 and YAP1 and concluded a key function for miR-30e-5p and miR-15a in mediating the differentiation of adipocytes, suggesting a role for them in stimulating the synthesis of milk fat [124].

When miR-126-3p is suppressed, the expression of the FASN gene increases, indicating that it is involved in lipid metabolism in the mammary gland. In addition, estradiol and progesterone enhance lipid synthesis by decreasing miR-126-3p levels [125]. Similarly, miR-150 inhibits lipogenesis in mammary epithelial cells and decreases secretory activity [126], whereas miR-145 facilitates milk-fat synthesis in lactating goats by targeting the INSIG1 gene (insulin-induced gene 1) [65]. miR-15b also suppresses the lipid metabolism in the mammary epithelial cells of goats and mice as FASN levels decrease [83]. The overexpression of miR-34b also suppresses the expression of mRNAs associated with FASN, FABP4 (fatty acid binding protein 4), and C/EBPα (CCAAT enhancer-binding proteins) [127]. In their previously mentioned work, Shen et al. (2016) selected three miRNAs that could be regulators of milk-fat metabolism by affecting their putative target genes: miR-33a, which is predicted to target ELOVL5, ELOVL6, and SC4MOL; miR-152, which is predicted to target PTGS2, PRKAA1, and CUP3; miR-224, which is predicted to target LPL, GST, ALOX15, and PTGS1 [128].

Further studies have shown that miR-145 can alter lipogenesis in goat mammary epithelial cells by targeting INSIG1 and other lipid-related genes [65]; miR-29s can regulate epigenetic changes in the lactation genes associated with casein alpha-S1 (CSN1S1), E74-like factor 5 (ElF5) (activated by PPAR-γ), SREBP1, and glucose transporter 1 (GLUT1) toinfluence milk-lipid metabolism [129]; miR-25 can also inhibit lipid synthesis in goat mammary epithelial cells through peroxisome proliferation-activated receptor gamma coactivator 1 alpha (PGC-1beta), which, in turn, promotes higher SREBP family activity [130]. miR-130b, meanwhile, directly represses the coactivator PPARγ-1α, thereby suppressing the fat metabolism and adipogenesis in goat mammary epithelial cells [131]; miR-454 and miR-34b perform the same function and according to the same principle in the epithelial cells of the mammary glands of cattle, also reducing fat-droplet accumulation [127,132]. miR-24 regulates fatty acid synthase genes, modulating the synthesis of triacylglycerol in goat mammary epithelial cells [133]; miR-27a also controls triacylglycerol synthesis in the mammary epithelial cells of cattle, reducing triacylglycerol accumulation, inhibiting lipid-droplet formation, and decreasing the unsaturated/saturated fatty acid ratio in mammary epithelial cells upon overexpression [85] by acting on the peroxisome proliferator activated gamma receptor [134], while itself being regulated by the circular RNA circ09863 [134].

The overexpression of miR-148a and miR-17-5p, in turn, promotes triacylglycerol synthesis, making these miRNAs key elements in this process, and cooperating with one another, they are able to synergistically repress the PPARGC1A and PPARA genes responsible for the fat metabolism and fatty acid oxidation, respectively, in goat mammary epithelial cells [81]. By modulating the insulin receptor substrate 2 (IRS2), miR-181b, in turn, is able to suppress the synthesis of triacylglycerol, and its overexpression disrupts fat metabolism [82], while miR-181a can control milk-fat biosynthesis by modulating acyl-CoA synthetase long chain family member 1 (ACSL1), which is an important enzyme for milk lipid formation [135]. miR-142-5P can also promote milk-fat metabolism by inhibiting the expression of catenin beta-1 (CTNNB1) [136]. Chu et al. (2018) found in their study that miR-221 could also regulate the lipid metabolism in mouse mammary epithelial cells by targeting the genes associated with lipid synthesis, namely FASN, ACSL1, ElF5, and NR1H3, and its expression decreased during lactation [137]. miR-143 significantly promotes lipid-droplet formation and increases the level of intracellular triglycerides due to an increase in the expression of genes associated with lipid synthesis, such as PPAR-gamma, FASN, SCD1, CEBPβ, and SREBP1, targeting Smad3 [138].

According to recent data, lipid synthesis, along with αs1-/β-casein formation, is also regulated by miR-8516, which is one of the elements of the circ-140/miR-8516/STC1–MMP1 (scenically-1–matrix metallopeptidase-1) feedback loop [139]. The inhibition of miR-183, which regulates the MST1 gene, also promotes milk-fat metabolism in goat mammary epithelial cells [140]. According to Jiao et al. (2020), another target for miR-183 is the IRS1 gene, which plays an important role in regulating milk-fat metabolism in the mammary epithelial cells of cows [96]. Shen et al. (2019) found that the overexpression or suppression of miR-124a in dairy cows resulted in the suppression or upregulation of the PECR gene (peroxisomal NADPH-specific trans-2-enoyl-CoA reductase), suggesting that miR-124a can target the 3’-UTR of the PECR gene, regulating its expression. PECR is involved in fatty acid metabolism, suggesting that miR-124a may indirectly influence milk-fat synthesis by regulating PECR. In addition, Shen et al. (2019) noted an increased triglyceride content when miR-124a was overexpressed in mammary epithelial cells, as well as an increase in free fatty acids [55]. Recently, the ability of miR-485 to regulate the synthesis of triglycerides, non-esterified fatty acids, and cholesterol in the mammary epithelial cells of cows, by inhibiting the expression of the DTX4 gene, which is a PPAR-γ transcription factor, was observed [141]. Additionally, Fan et al. (2021) recently demonstrated the regulation of the synthesis of polyunsaturated fatty acids by miR-193a-5p through the modulation of fatty acid desaturase 1 (FADS1), leading the authors to conclude a role for this miRNA in regulating the amount of milk fatty acids in the epithelial cells of the mammary glands of cattle [142].

Determining the epigenetic aspects of lipid metabolism in the mammary gland, particularly in lactogenesis, is playing an increasingly crucial role in elucidating the importance of miRNAs in this process. The quantitative predominance of studies in this direction conducted in the last few years suggests that this issue has received proper attention quite recently and will be the subject of many new discoveries in the future, which will help in compiling a unified picture of the participation of miRNAs in the regulation of fat metabolism in the mammary gland. This is also demonstrated in Table 6.

### 3.4. Mammary Gland Cells

The mammary gland is a unique and dynamic organ that exhibits the many stages of the female reproductive cycle. Between the beginning of pregnancy and breastfeeding, the mammary gland undergoes extensive reconstruction; this implies cellular processes such as cell proliferation, differentiation, and apoptosis, all of which are under the control of multiple regulators. As explained in the section on the hormonal regulation of miRNAs in the mammary gland, during mammary gland differentiation, lactogenic hormones such as prolactin and glucocorticoid stop the growth of mammary epithelial cells and initiate the production of milk protein [143]. Because lactating mammary glands synthesize more proteins than other organs, precise gene regulation is important for coordinating cellular and tissue remodeling during the stages of differentiation.

miRNAs are also known to be involved in the regulation of various cellular processes [144]. Given that the majority of miRNAs identified to date regulate critical cellular processes, including proliferation, differentiation, and apoptosis, it is fair to anticipate that many miRNAs are involved in mammary gland development [145]. Many studies have shown that miRNAs control mammary gland development by regulating the formation of mammary ducts and acinus [146], as well as the proliferation and apoptosis of mammary epithelial cells [35]. The dairy yield of milk strongly depends on the number of epithelial cells and their secretory activity [147]. In this regard, it is important to establish the effects of miRNAs on mammary epithelial cells.

For example, Nagaoka et al. (2013) analyzed the miRNA micromatrix during the differentiation of mammary epithelial cells in mice and found that miR-101a could regulate cell proliferation by targeting COX-2 expression, which can play an important role in mammary gland differentiation and involution [148]. Later, to better understand the importance of miR-200a during mammary gland development, the importance of miR-200a in epithelial cell differentiation and cell polarity was confirmed in miR-200a knockdown experiments, demonstrating its role in controlling E-cadherin [149]. In many studies in this direction, particularly the previously mentioned work by Galio et al. (2013) studying temporal miRNA patterns in the sheep mammary gland, it has been suggested that miR-200a plays a critical role in maintaining the epithelial-cell phenotype [76,149,150]. According to the assumption of Galio et al. (2013), miR-200a is involved synergistically along with miR-205. Greene et al. (2010) also focused their research on miR-205, which, as was established, is the most highly expressed miRNA in the mammary epithelial cell precursor population of Sca-1 mice. The overexpression of miR-205 leads to an increase in the progenitor cell population, a decrease in cell size, and an increase in cell proliferation [151]. High levels of miR-205 have also been observed in normal lobular and ductal myoepithelial cells of human mammary glands, indicating its potential role in their normal development [152].

miR-24-3p negatively regulates the expression of the MEN1 gene and a protein that inhibits epithelial cell proliferation, menin, which is involved in the regulation of mammary gland development; the miRNA also modulates the expression of prolactin’s mRNA by binding to its promoter and, therefore, indirectly regulating mammary cell development [153], as well as modulating the proliferation of mammary epithelial cells through the cell-cycle regulator cyclinD1 [154]. However, miR-24-3p expression, in turn, is also controlled by the product of MEN1 expression, indicating a negative feedback loop between miR-24-3p and MEN1. Jiao et al. (2019) found that the overexpression of miR-221 in cow mammary epithelial cell cultures reduced cell viability and inhibited cell proliferation. To elucidate the molecular mechanisms of miR-221’s effects on cell proliferation, they selected potential candidate genes that could target miR-221 using bioinformatic prediction tools. Dual luciferase assays showed that the STAT5a, STAT3, and JAK–STAT signaling pathway members play a key role in the proliferation, secretory differentiation, and survival of breast epithelial cells [155], along with IRS1. The type 1 insulin-like growth factor receptor (*IGF1R*) and insulin receptor (IR), which play a central role in cell signaling molecule networks, including the PI3K–Akt/mTOR signaling pathway [156], interact with miR-221 by directly binding to the 3’-untranslated regions (3′-UTRs) of these genes. Subsequent analysis showed that miR-221 transfection led to a significant decrease in STAT5a and IRS1 expression at both the RNA and protein levels [80]. Ji et al. (2019) studied the regulation of the protein interacting with the Nedd4 1 family (Ndfip1), which plays one of the leading roles in the ubiquitination process [157], at the post-transcriptional level to determine its participation in the development of mammary gland cells. Upon analyzing the expression of Ndfip1 at the mRNA and protein levels by qRT-PCR and Western blotting, respectively, after the overexpression and inhibition of miR-143-3p, it was shown that this miRNA regulated the expression of Ndfip1 in mammary goat epithelial cells cultured in vitro, and using flow cytometry, the authors were able to prove that miR-143-3p’s modulation of the expression of Ndfip1 increased the numbers of early, late, and completely apoptotic cells [158]. Chen et al. (2019) found that the overexpression of miR-145 in cow mammary epithelial cells significantly suppressed FSCN1, thereby reducing interleukin-12 secretion and tumor necrosis alpha, but increasing interferon gamma secretion, which, in turn, based on their assumption, suppresses cell proliferation [159].

At the same time, a unique and interesting aspect of mammary gland biology is the need for mammary epithelial cells to grow and function in a stromal environment called the mammary fat pad [160]. Resident adipocytes are crucial for this environment. In this regard, it is relevant to pay no less attention to the influence of miRNAs on adipose tissue formation and on adipogenesis in particular. Guo et al.’s (2012) results demonstrate that the increased expression of miR-145 inhibits adipogenesis by modulating the insulin receptor substrate 1, which distinguishes it from most miRNAs, including miR-210 [161], miR-217 [91], miR-103/107 [162], and the miR-17-92 cluster [163]. Moreover, according to their information, miR-145 and miR-143, which are in the same cluster as those miRNAs, played completely opposite roles in adipogenesis because of their different native sequences [164]. Yes-associated protein (YAP) and its downstream proteins in the Hippo signaling pathway are known to play a crucial role in stimulating adipocyte growth and inhibiting apoptosis [165]. Regarding adipocytes in mammary tissues, in the previously cited study, Chen et al. (2016) demonstrated a synergistic suppressive effect of miR-15a on YAP1 via β-catenin in the mammary glands of goats [124]. A link between adipocyte formation and miR-130 expression has also been established, as described in the previous section about reduced fat accumulation in cells. Lee et al. (2011) indicated that miR-130 reduces PPAR-α expression and, consequently, adipocyte differentiation in humans [166]. Later, Chen et al. (2015) stated that miR-130b overexpression also disrupted adipogenesis in the mammary epithelial cells of goats [131]. Liang et al. showed that miR-25 was suppressed during adipocyte differentiation and suppressed 3T3-L1 adipogenesis by modulating Kruppel-like factor 4 and CCAAT/binding enhancer protein alpha [167].

Thus, the involvement of miRNAs in the regulation of the mammary tissue development of farm animals has received no less attention from researchers than other aspects. For clarity, the essence of miRNAs’ function in this regard is shown in the Table 7.

## 4. Dependence of Mammary miRNA Expression on External Conditions

### 4.1. Immunological Role of Mammary Gland miRNAs and Their Functions in Breast Diseases

Accumulating evidence suggests that miRNAs are an important part of the complex regulatory networks that control cellular processes, including those of immune cells. The immune responses at different stages of the innate immune network, including cytokine/chemokine production and release, the expression of adhesion and costimulatory molecules, and the transport of miRNAs via the release of exosomes and feedback regulation of immune homeostasis, are modulated by miRNAs [168]. Studies have shown that, after cells receive exogenous or endogenous signals, changes in miRNA expression are part of the early response. For example, the expression levels of certain miRNAs change rapidly after the onset of a disease in the body to control the extent and intensity of the disease [169,170,171]. Thus, miRNA expression levels during disease onset or progression can be used as an early diagnostic biomarker of the disease [170,172]. There is ample evidence that miRNAs play a fundamental role in the development and function of both innate and adaptive immune cells [173,174]. For instance, miR-21, miR-146a, and miR-155 are well-studied miRNAs involved in immune responses initiated by Toll-like receptor signaling pathways and regulate nuclear transcription factor kappa B (NF-kB) activation [175]. It has also been suggested that some miRNAs play an immunosuppressive role. The in silico analysis of two members of the miR-30 family (miR-30a-5p and -30d-5p) has predicted binding sites in several suppressors of cytokine signal-transduction inhibitors of the JAK/STAT pathway that regulate IL-10 transcription [176]. These miRNAs are also involved in the formation of the fat pad in the mammary gland. Breast milk is enriched with miRNAs, such as miR-22-3P, which regulate the development and differentiation of T-lymphocytes, and miR-181a-5p and miR-182-5p play a crucial role in the differentiation of immune cells [177].

In recent years, various studies have shown that bovine mammary epithelial cells respond to the invasion of bacteria or their waste metabolic products by changing the expression levels of certain genes involved in inflammatory processes and immunity in vitro [178,179,180]. For example, it was recently found that the expression of miR-27a-3p in mammary epithelial cell cultures from dairy cows is significantly increased by the administration of lipopolysaccharide, which is a key antigenic factor for bacterial mastitis pathogens [181]. Mastitis is the most common inflammatory udder disease of dairy cattle, usually caused by bacterial infection [182]. miRNAs related to bovine mastitis have been isolated from the peripheral blood [183], milk exosomes [184], breast tissue biopsies [185], and breast epithelial cells [59] of infected animals. Several studies of the epigenetic mechanisms of the regulation of the immune response to mastitis have used a comprehensive approach based on the integrative analysis of miRNA and mRNA expression profiles to improve the understanding of the underlying molecular mechanism of mastitis in cattle caused by *S. aureus*. Thus, in order to investigate the different interaction networks and modes of mRNA and miRNA regulation, Wang et al. (2021) constructed a model of *S. aureus* bovine mastitis and integrated miRNA and mRNA analysis for cows infected with *S. aureus* and controls. They identified 77 differentially expressed miRNAs, and through integration analysis, they found that miR-19b, miR-23b-3p, miR-331-5p, miR-664b, and miR-2431-3p were potential factors regulating the expression levels of G-protein subunit gamma 2 (GNG2), member 2 of the RP/EB protein family associated with microtubules (MAPRE2), CD14 molecule (CD14), interleukin 17A (IL17A), calcium-binding protein S100 A9 (S100A9), type IV collagen alpha-1 chain (COL4A1), a member of the RAS oncogene family (RAP1B), NFKB signaling regulator LDOC1 (LDOC1), and low-density lipoprotein receptor (LDLR), which are generally associated with inflammation and immunity [186,187,188,189,190,191].

Lawless et al. (2014) found three miRNAs (bt-miR-146B, -451, and -411a) in the blood monocytes of dairy cows infected with *S*. *uberis* in vivo [192]. Lee et al. and Chen et al. used qRT-PCR and a Genome Analyzer IIe (Illumina) to detect miRNAs in the peripheral blood of dairy cows affected with mastitis [183,193]. Their findings revealed that, compared to in healthy cows, there were 123 miRNAs linked with mastitis that exhibited substantial upregulation (e.g., miR-15a, miR-16a, miR-21-3p, miR-29b, miR-125b, and miR-181a, miR-148a, miR-223, miR-375, and let-7f). However, it is remarkable that the findings of the two authors varied significantly, with some findings seemingly contradicting others. For example, the results of Li et al. and Chen et al. show that miR-146a and miR-146b were significantly suppressed in the blood of cows with mastitis in one control group [193] but significantly activated in the other control group [183]. Thus, it is reasonable to assume that the specific population of dairy cows, the type of infection, and the time of infection (or the degree of mastitis) have a significant impact on the miRNA expression profiles in the blood. Therefore, Zhuo-Ma et al. (2018) used a small dose of *E. coli* for the long-term infection of mammary gland tissue and used RNAseq technology to detect miRNA expression in the blood of dairy cows at different times after infection. In total, they discovered 200 differentially expressed miRNAs (including 76 known and 124 new miRNAs), of which the expression of miR-200a, miR-205, miR-122, and miR-182 was significantly elevated in late mastitis, while conservative_15_7229, the functions of which still have not been reported, was significantly suppressed. Their findings also revealed that these miRNAs can participate in seven signaling pathways associated with animal immunity, of which the Toll-like receptor signaling pathway [194] and the chemokine signaling pathway are closely linked to innate immunity and inflammatory responses [195], whereas the T-cell receptor signaling pathway [196] is associated with adaptive immunity, which allows one to conclude that these miRNAs may be involved in innate or adaptive immunity in mastitis of dairy cows and, thus, can regulate the development of mastitis [197].

Lai et al. (2021) analyzed the expression levels of miR-21, miR-146a, miR-155, miR-222, and miR-383, and using a digital PCR system, they found that only the expression level of miR-21 significantly increased in the serum of the blood of cows with mastitis compared to that of healthy cows. Increased expression of miR-21 in the blood was also observed by Chen et al. (2014) in cows infected with *S. aureus* [193]. Jin et al. (2014), in turn, observed increased expression of miR-21-3p 24 h after *S. aureus* infection [59]. Similar data were obtained by Fang et al. (2016) [198]. miR-21-3p has been suggested to negatively regulate the vitamin D-dependent antimicrobial pathway during the infection of *Mycobacterium leprae* [198,199], which is biologically important for the innate immune system response to infection and wounds, and its deficiency leads to a suboptimal response to bacterial infection [200]. CALB, the target gene for miR-21-3p, according to the study of Fang et al., is involved in the biological process of vitamin D binding, which may indicate that miR-21-3p affects the vitamin D-dependent antimicrobial pathway, in part, through the post-transcriptional downregulation of CALB1 [201,202]. Exosomes originating from sensory neurons containing miR-21-5p can increase the expression of proinflammatory genes and proteins during macrophage phagocytosis [203]. This suggests that increased expression of circulating miR-21 in cows affected by mastitis may play a role in communication with other system bodies [204].

Of the 48 miRNAs detected by Han et al. (2020), miR-223 and miR-21-5p were most noticeably expressed, whereas miR-205 showed a marked decrease in mastitis, which was not observed in healthy breast tissues. Meanwhile, miR-223, according to their observations, suppressed the inflammatory response in lymphotoxin-stimulated mammary alpha T cells by acting on the protooncogene B Cbl (CBLB) [62]. Another study showed that miR-223 was activated and miR-205 was suppressed in the mammary glands of cows infected with *S. aureus* [185]. In addition, miR-223, miR-9, miR-125b, miR-155, and miR-146a were highly expressed in bovine CD14^+^ monocytes stimulated by lipopolysaccharide or enterotoxin B derived from *S. aureus* [205]. Using real-time quantitative PCR, Western blotting, and luciferase multiplexing verification methods, as mentioned earlier, Chen et al. (2019) found that, in breast tissue infected with *S. aureus*, miR-145 expression was suppressed, thereby reducing its suppression of the FSCN1 gene. Increased expression of FSCN1 promotes the proliferation of mammary gland epithelial cells and increases the levels of the cellular immune cytokines secreted by these cells [159]. A little earlier, the same authors, using RT-PCR, revealed the expression of miR-196a, miR-205, miR-200b, miR-31, miR-145, miR-223, miR-184, and miR-132 in mastitis-affected mammary glands in cows, suggesting that these differentially expressed miRNAs could be used as markers of mastitis caused by *S. aureus*. In addition, their luciferase-reporter assay results indicate that miR-15a significantly inhibited luciferase activity, suggesting that it acted through the 3’-UTR of IRAK2 to suppress reporter-gene expression. Studies have shown that IRAK2 has a significant negative regulatory function and can reduce the mast-cell apoptosis induced by lipopolysaccharide infusion; miR-15a, according to previously obtained data, is involved in multiple functions that regulate the inflammation and differentiation of immune cells [206,207,208].

Ju et al. (2018), using quantitative real-time PCR, in situ hybridization, and a dual luciferase reporter assay, determined the effect of miR-26a in mastitis, mediated, at least in part, by the increased expression of fibrogen alpha (FGA). The interaction between miR-26a and FGA reduced the activity of the luciferase reporter gene, indicating that FGA is one of the direct target genes for miR-26a. Thus, they suggested that miRNA-mediated regulation could be a potential mechanism for the inconsistent levels of FGA protein expression after recurrent *S. aureus* infection in cows [58]. Interestingly, the activity of FGA, a binding molecule, is probably partially mediated by *S. aureus* binding to platelets, which is an important host defense mechanism against mastitis infection [209]. In addition, four of the 10 miRNAs they found expressed in the mammary gland, namely, miR21-5p, let-7f, let-7a-5p, and miR-148a, were similar to the 10 miRNAs expressed in the epithelial cells of cattle mammary glands with or without infection by heat-inactivated *E. coli* or *S. aureus* [210]. Notably, some of the most highly expressed miRNAs found in this study have been shown to be responsible for the formation of immunity. For example, the most widely expressed miRNA, miR-143, may be associated with marked suppression of the gene expression of BCL2, an apoptosis regulator, during the inflammation induced by bacterial lipopolysaccharides, and could additionally be involved in suppressing the translation of several other predicted inflammation-related target genes [211]. Another case is miR-21, a negative feedback regulator that suppresses the immune response that is induced by NF-κB during the TLR4 induction of macrophages in a MyD88-dependent manner. After its induction, miR-21 targets mRNA encoding programmed cell death 4 (PDCD4), a proinflammatory tumor-suppressor protein that activates NF-κB by currently unknown mechanisms. This effect leads to the suppression of NF-κB signaling and activation of the anti-inflammatory response, depending on IL-10 secretion [212].

Five miRNAs, namely, let-7f, let-7a-5p, let-7b, let-7c, and let-7g, belong to the let-7 family, whose role in the regulation of innate immunity has also been described in detail in the literature [213]. Ju et al. (2018) also constructed a network between differentially expressed miRNAs and their target genes and found several other immunity-related target genes. Interestingly, the activated leukocyte adhesion molecule (ALCAM) gene, a target of miR-148a, also known as the CD166 antigen gene, encodes a transmembrane receptor that is involved in leukocyte adhesion and migration, as well as in T-cell activation [58]. It has been shown that ALCAM is overexpressed in the somatic milk cells of a mastitis-resistant sheep line compared to a mastitis-sensitive line [214]. Therefore, it should be considered a probable candidate gene involved in bovine mastitis resistance in the framework of a genome-wide association study [215]. m0075-5p, newly discovered by Ju et al. (2018) as a downregulated miRNA, presumably targets an important gene with high mobility in group 1 (HMGB1), which performs a universal signaling function in nucleic acid-mediated innate immune responses to bacterial infection and acts as a pathogenic mediator in inflammatory diseases [58]. Previously, they also found that HMGB1 was modulated by miR-223 and an SNP in HMGB1′s 3′-UTR, and a change in binding between HMGB1 and miR-223 is associated with changes in somatic cell counts in cows [216]. It has also been confirmed that miR-2898, which showed the highest differential expression in this study, is also activated 6.25 times more intensely in the mammary gland tissues of cows infected with mastitis [217]. miR-2898 can target a host-binding protein—A2M—or foreign peptides and particles, protecting against pathogens in animal plasma and tissues [217]. The authors of that study identified 14 proteins, including fibrinogen alpha chain (FGA), fibrinogen beta chain (FGB), collagen type I alpha 1 chain (COL1A1), inter-alpha trypsin inhibitor heavy chain 4 (ITIH4), and C-reactive protein (CRP), which are targeted by their potential miRNAs, and two proteins, complement C3 (C3) and calcium-binding protein A12 S100 (S100A12), which have no corresponding miRNA targets. They also found that the let-7 miRNA family (let-7a, let-7b, let-7c, let-7d, let-7e, let-7f, btalet-7g, and let-7i) can synergistically enhance the expression of the calcium-binding protein S100 A8 (S100A8) and proteins COL1A1 and ITIH4, which are associated with immunity, inflammation, defense responses, tissue damage, and tissue repair in mastitis cows in the late stages of *S. aureus* infection [209]. Three activated miRNAs, namely, miR-2900, miR-2360, and miR-2449, presumably suppress the expression of apolipoprotein A4 (APOA4), which is a multifunctional protein involved in several GO terms and pathways, including immune system processes (GO: 002376), adaptive immunity (GO: 0002250), and innate immunity (GO: 0045087). Thirteen activated miRNAs, such as miR-2442, presumably target C-reactive protein (CRP), a nonclonal host-resistance effector [218].

It is also important to note that the immune responses of bovine mammary epithelial cells infected with *E. coli* and *S. aureus* differ significantly [219,220]. In addition, the differential expression of inflammatory cytokines and other immune-related proteins in response to *E. coli* or *S. aureus* has also been observed in breast tissue and milk [221,222,223], and several studies of transcriptomic differences in the mammary glands of cattle infected with *S. aureus* or *E. coli* have been reported. The data obtained by Luoreng et al. (2018) allowed the detection of differentially expressed miRNAs by a pairwise comparison of *S. aureus*- and *E. coli*-infected and healthy cows [60]. Although differentially expressed miRNAs have previously been reported in breast tissue infected with a high dose of *S. aureus* [185], the results differ slightly in terms of the amounts and types of miRNAs, suggesting that the dose of the bacterial inoculum or the duration of the infection may affect miRNA expression in the mammary glands. The miRNA expression patterns in the control and *S. aureus*-infected groups were based on a total of 279 miRNAs, including 186 activated and 93 suppressed miRNAs. In addition, a total of 305 miRNAs were identified in samples infected with *E. coli*, including 243 upregulated and 62 downregulated miRNAs. Among these miRNAs, it seems that miR-7863 may be a specific biomarker for the two types of mastitis studied here, due to the fact that the expression in the *S. aureus*- and *E. coli*-infected groups was increased 24 times compared to that in the control. The potential target genes predicted by bioinformatic analysis showed that miR-7863 could regulate several immune genes, including members of the interleukin family, IRAK1, TLR7, and LBP, which are the key immune system molecules of many animals. Moreover, miRNAs with high expression in the control and in the group infected with *S. aureus*, including miR-223, miR-146a, miR-184, miR-155, miR-214, miR-147, and miR-378, are also differentially expressed [60]. The expression levels of many miRNAs characterized in this study are in line with those reported in other early studies. For example, a previous study by Luoreng et al. (2018) showed that miR-146a expression in bovine mammary gland tissues was significantly increased in cases of mastitis caused by *S. aureus*, *E. coli*, or mixed bacterial infection [224], supporting the idea that bovine miR-146a regulates the secretion of inflammatory cytokines such as TNF-α, IL-6, and IL-8 in bMEC [225]. It is also noteworthy that miR-375 is one of the most suppressed miRNAs, which allowed the authors to suggest that it is involved in the regulation of immune responses and inflammation. Moreover, they observed that the changes in the expression of miR-144 and miR-451 in the two groups were opposite to one another (upregulation in the mammary gland infected with *S. aureus* and downregulation in the mammary gland infected with *E. coli*), suggesting that they play different roles in the mechanisms regulating the two types of mastitis they explored. To investigate their roles, the potential target genes were predicted using bioinformatic analysis, showing that miR-451 could regulate ATF2, which, in turn, controls the levels of cytokines, CDKN2D, and MEF2D, which is involved in the regulation of inflammatory responses [226,227].

For miR-144, a total of 31 potential target genes involved in immunity, such as EZH2 and NKRF, were predicted. Several studies have reported that the role of EZH2 lies in the regulation of T cell differentiation and function [228], as well as in the negative regulation of immune responses [229]. Moreover, NKRF suppresses NF-kB activity and subsequently regulates the NF-kB signaling pathway, which is a key signaling pathway of the innate immune response [230,231]. Additionally, in their study, the putative miRNA target genes were annotated as being involved in various classes of immune signaling pathways, the TLR signaling pathway, the TGF-β signaling pathway, cytokine–cytokine-receptor interactions, the MAPK signaling pathway, cell-adhesion molecules, transendothelial leukocyte migration, and chemokine signaling pathways [60]. It has been previously reported that all of these signaling pathways may be associated with the development of mastitis [201,219,232,233].

Previously, Wang et al. (2017) investigated the miRNA expression profiles of cattle mammary glands in response to mastitis caused by *Streptococcus agalactiae*. In the test group, 35 differentially expressed miRNAs were identified compared with the control group, including 10 overexpressed miRNAs and 25 downregulated miRNAs. Of these miRNAs, miR-223 exhibited the highest degree of upregulation, with an approximately 3-fold increase in expression. In previous studies of bovine mastitis, miR-223 was also activated 2–5 fold [234]. A significant increase in the miR-223 level with a high bacterial load of *S. aureus* was also observed by the aforementioned Fang et al. (2016) [201]. miR-223 plays a key role in inflammation; for example, this miRNA negatively regulates the proliferation and differentiation of neutrophil granulocytes and reduces the increased expression of the factor E2F1, limiting cell-cycle progression [235,236]. miR-223 also inhibits several signaling pathways modulating IGR1R [234]. Validated miR-223 targets, including GZMB, IKKa, RC3H1, and STAT3, impact inflammation and infection [237]. Research by Wang et al. (2017) showed that some of the putative miR-223 target genes are involved in the immune pathway during *S. agalactiae* mammary tissue infection. They identified that miR-16, similarly to miR-223, was also related to the immune response. Previous studies have shown that miR-16a activates the interleukins IL-6, IL-8, and IL-10 after the infection of cows with *S. uberis* [234]. miR-16, as mentioned earlier, modulates macrophage polarization, inflammasomes, and NF-kB signaling [193]. miR-136 has also been detected as probably targeting CD93, which is a membrane-associated glycoprotein on the cell surface that mediates phagocytosis, inflammation, and cell adhesion [238].

It has been found that breastmilk miRNAs are more sensitive biomarkers of breast diseases than blood miRNAs [239]. In this regard, Lai et al. (2017) analyzed cow’s milk for mastitis and found that the expression levels of miR-155, miR-146a, miR-222, miR-383, and miR-21 are significantly increased in milk positive for mastitis [240]. They later identified 25 differentially expressed miRNAs in milk from infected cows, including miR-1246, miR-223, miR-142-3p, miR-142-5p, miR-21-3p, miR-6529, miR-147, miR-505, miR-2284aa, miR-2284w, miR-132, and miR-130b, which were upregulated, as well as miR-874 and miR-23b-3p, which were suppressed [175]. These miRNAs are known to regulate the expression of immune-related genes, including CXCL14 and KIT, suggesting that they play roles in mammary gland post-transcriptional responses in *S. aureus*-inducted mastitis [201]. miR-10a, miR-146a, miR-146b, miR-221, and miR-223 are associated with the regulation of innate immunity and mammary epithelial cell functions in *S. uberis*-infected tissues [234]. miR-146b, miR-223, and miR-338 were shown to be activated in monocytes isolated from milk and blood after *S. uberis* infection [192]. Meanwhile, the expression of the miR-30 family, including miR-30f, was significantly suppressed, while miR-222 was significantly correlated with the number of somatic cells, suggesting the utility of miR-222 as an indicator of mastitis in studies of milk samples containing fat and somatic cells (without centrifugation) [192]. miR-301a can activate NF-κB signaling and has been shown to be suppressed in the blood of mastitis-affected cows. Chen et al. (2014) found a potential role for miR-29b-2 as a biomarker for identifying mastitis-infected milk [193]. Sun et al. (2015) hypothesized the presence of specific miRNAs in milk exosomes, the presence or absence of which can serve as a biomarker for the early detection of bacterial infection that can lead to mastitis. In this regard, by the deep sequencing and generation of sequence reads from unconjugated miRNA libraries, they analyzed and compared the miRNA expression profiles of milk exosomes from four Holstein cows obtained in mid-lactation before and after *S. aureus* mammary infection and identified six miRNAs significantly differentially presented in exosomes in response to bacterial mammary infection: miR-101, miR-142-5p, miR-183, miR-2285g-3p, miR-223, and miR-99a-5p. They observed increased levels of miR-142-5p and miR-223 in milk exosomes 48 h after infection [184]. Other studies of cow’s milk have discovered greater levels of miR-223 in colostrum, which might promote enhanced immunity in infants or mammary glands during a period of higher (postpartum) vulnerability to bacterial infection [33,49]. miR-142 has been shown to be abundantly present in T cells, implying a role for it as an immune-relevant miRNA [241]. miR-223, as described above, plays a wide role in balancing the metabolism and immune response during infection [192,234]. Sun et al. also observed decreased levels of miR-15b and miR-193a-5p, found only through SOAP analysis, which is confirmed by similar data in the mammary gland epithelium [184,234].

Thus, the identification of specific genes associated with susceptibility or resistance to mastitis may provide a new way to combat mastitis through genetic selection [242,243]. It should be noted that some miRNAs can serve as universal biomarkers of mastitis, regardless of the causative agent, such as miR-223 and miR-146, listed in this section, as well as the miR-21 and let-7 families. In Table 8, we reflect how mammary gland miRNAs can be associated with the health and diseases of this organ.

### 4.2. Heat Stress

Heat stress is caused by an increased ambient temperature, which occurs when the body’s temperature exceeds the upper threshold of the thermoneutral zone [244]. Ruminants exhibit different responses to acute and chronic heat stress, such as increased protein catabolism and decreased protein synthesis; at the same time, chronic heat stress decreases catabolism and protein synthesis [245,246], which leads to a decrease in milk yield and the contents of protein, lactose, and fat in milk [247], as well as immunity [248] and reproductive capacity [249]. The studies conducted to date have identified some physiological, metabolic, cellular, and molecular responses to heat stress, and recently, much attention has been paid to the effects of heat stress at the genetic level, that is, at the gene-expression level [37,250,251,252]. Since it became clear, through the study of epigenetics, that the response to heat stress is a complex molecular process that includes both the transcriptional and post-transcriptional regulation of genes, there was interest in investigating the effect of heat stress on an epigenetic element such as miRNAs, which also occurs in cells of the mammary gland, which is important for better understanding the role of miRNAs in the regulation of mammary gland functions.

Liu et al. (2020) suggested that miR-423-5p may be an important regulator of the response to heat stress in cows, based on the increased expression of this miRNA under heat stress [253]. As for the direct epigenetic responses to heat stress in the mammary gland, miR-216b has been found to inhibit the heat-stress-induced apoptosis of mammary epithelial cells under heat stress [254]. It has also been observed that miR-216 is more strongly expressed in cows with an increase in not only temperature but also air humidity [255]. In the previously mentioned study by Li et al. (2018), in addition to finding 139 new miRNAs, the epigenetic responses to heat stress in cows were also studied, and it was observed that the expression of miR-145, miR-133a, and miR-29c was increased in the heat-stress group. The expression of miR-2285 and miR-146b decreased in the heat-stressed group compared to the control, which is also confirmed by the observation of Zheng et al. (2014) [256]. They also found that the expression of miR-21-5p and miR-146b tended to decrease, while miR-145 tended to increase in the heat-stress group compared to the control group, which may indicate that heat stress suppresses lncRNA, which, in turn, inhibits the action of miR-145 [257]. The fact that acute heat stress affects lipolysis and the rate-limiting enzyme of lipogenesis in bovine adipocytes is supported by these findings [37,258]. Previously, the aforementioned target gene analysis of Zheng et al. (2014) found that at least eight miRNAs (miR-19a, miR-19b, miR-27b, miR-30a-5p, miR-181a, miR-181b, miR-345-3p, and miR-1246) were involved in the response to heat stress [256]. Conducting a study on 12 cows, Fan et al. (2020) found 124 miRNAs both positively and negatively correlated with heat-stress stimulation, such as let-7c, let-7e, miR-181d, miR-452, and miR-31, which share the same target gene, IL-1, whose upregulation not only causes a systemic inflammatory response, but also impairs the health and physiological functions of dairy cows by altering blood sugar [259,260]. Thus, they concluded that these differentially expressed miRNAs may be involved in the development of udder inflammation under heat stress and, in contrast to the phenomenon observed by Cai et al. (2018), promote the apoptosis of mammary gland cells, which ultimately affects milk composition and milk yield [261].

Thus, we can conclude that the study of the stress mechanism and stress-sensitive indicators, as well as the processes regulating them, particularly epigenetics, are key for reducing the damage caused by heat stress.

### 4.3. Food Components and Additives

As established, the production and composition of ruminant milk are associated with both internal and external factors, such as nutrition [262]. Numerous genes are involved in the synthesis and secretion of milk components by mammary epithelial cells. Interestingly, nutrition affects the expression of genes encoding important milk production factors in ruminants, which ultimately affects milk quality [262,263,264,265,266,267]. Thus, it has been shown that 48 h starvation of lactating goats causes the decreased production of milk and secretion of components, which is associated with altered expression of 161 genes, including those encoding lipogenic enzymes and basic milk proteins [263]. An increased level of unsaturated fatty acids in the diets of cows affected 972 genes associated with cell development and remodeling, apoptosis, the metabolism of nutrients, and immune system responses, which led to an increase in milk yield but decreased percentages of milk fat and protein [268].

Reportedly, the expression and function of miRNAs can also be modulated by under- or over-feeding diets [262,269]. For example, calorie restriction increases the expression of miR-140-3p in the rat epiphyseal growth plate, as determined by microarrays and RT-qPCR assays [270]. A six-month caloric restriction has a substantial influence on the mouse mammary miRNA profile, as assessed using a microarray, according to Ørom et al. [271]. In addition, the expression of approximately 15 known and predicted miRNAs is altered in the muscles of calorie-restricted monkeys compared to those of ad libitum-diet monkeys [262]. More recently, Billa et al. (2021) investigated the breast miRNome of cattle upon restricting feed by half for six days. In their experiment, the authors managed to achieve a negative energy balance and lower milk yields, as well as lower levels of fats and proteins in both breeds during the early lactation period, for which a negative energy balance is known to be characteristic [272], while at the post-transcriptional level, the expression of 374 mRNAs in the mammary glands of Holstein cows changed; however, no significant changes in miRNAs were observed in Montbeliarde meat-and-dairy cows [273]. It should be noted that the composition and diet of dairy cattle differ in different periods of reproductive development [274,275]. A comparison of miRNAs after inducing a negative energy balance in cows in this study and also during early lactation in goats revealed similarities in the miRNomes between these two species [276]. At the same time, Ji et al. observed 378 miRNAs in the mammary glands of goats between early and late lactation periods, using all the miRNA sequences of mammals for mapping [277].

Wang et al. (2012) showed that the expression of 13 miRNAs in MG cattle was higher under a negative energy balance (i.e., early lactation) than under a positive energy balance (i.e., dry period). Despite the differences in the physiological states of the animals in the studies of Wang et al. (an early period compared to a period without lactation) and Billa et al., the two differentially expressed miRNAs (miR-155 and miR-181a) are common and regulate the energy balance in comparable fashions [45,273]. In addition, the genome position analysis of 25 known miRNAs in the study of Billa et al. showed that eight miRNAs were associated with milk QTLs: miR-155 is within the QTL associated with the α-lactalbumin content in milk, and another seven miRNAs are associated with the milk-fat composition. miR-143, which promotes lipid-droplet formation and raises intracellular triglyceride levels in breastfeeding, is the most highly expressed miRNA under a negative energy balance [138,273]. The second and third most expressed miRNAs are miR-26b and miR-181a, respectively. Increased regulation of miR-26b may be associated with inflammation due to the restriction of energy derived from food [278]. miR-181a targets mRNA ACSL1, which participates in the lipid synthesis in epithelial cells of the mammary glands of cattle [135]. The observed suppression of miR-200b and miR-200c may also indicate a link with structural changes in MG due to power limitations [273,279]. A similar study was previously conducted by Mobuchon et al. (2015) on goats. Of the 30 miRNAs identified by high-throughput sequencing with expression potentially modulated by food restriction, 16 were reduced and 14 were elevated. The authors observed increased expression of miR-126-3p, let-7c-5p, miR-99a-5p, miR-125b-3p, miR-140-3p, miR-409-3p, miR-451-5p, miR-99a-3p, miR-188-5p, miR-196a-5p, miR-204-5p, miR-222-3p, miR-223-3p, miR-494-3p, miR-660-5p, and miR-6119-5p, while miR-99a-3p, miR-188-5p, miR-196a-5p, miR-204-5p, miR-222-3p, miR-223-3p, and miR-494-3p expression remained almost unchanged, and the expression of miR-223-5p, miR-541-5p, and miR-671-5p decreased [262]. miR-126-3p has been characterized as a modulator of the TGF pathway influencing epithelial-to-mesenchymal transition in normal mouse mammary glands, while miR-99a-5p has been described as a modulator of the TGF pathway affecting epithelial-to-mesenchymal transition in the mammary glands of normal mice [280]. The autophagosome formation markers MAP1LC3B2 and ATG7 have also been found to potentially target miR-188-5p and miR-223-3p, respectively. These observations coincide with the decline in milk productivity and component synthesis previously observed in goats [263]. LPIN2 (LiPIN 2), which has recently been discovered in the mammary glands of cattle [281] and is implicated in triacylglycerol buildup, can likewise be targeted by miR-204-5p [282]. miR-671-5p may target the FADS1 (fatty acid desaturase 1) gene, which is involved in triacylglycerol production [283]. Moreover, miR-125b-3p, miR-494-3p, and chr3_3319-5p can jointly target ABCA1 (a member of subfamily 1 of ATP-binding cassettes), which is supposed to be involved in cholesterol transport, storage, and elimination in the mammary gland [284]. Finally, five nutrient-regulated miRNAs, namely, miR-188-5p, miR-222-3p, miR-494-3p, miR-541-5p, and chr3_3319-5p, can target PTEN (phosphatase homologue and TENsin). Meanwhile, several genes encoding casein kinases 1 and 2 alpha and gamma may be targeted by miR-222-3p, miR-409-3p, miR-541-5p, miR-660-5p, miR-6119-5p, chr3 4386-5p, and chr23 30758-5p [262].

A number of other researchers have expressed interest in studying the effects of some food additives and general dietary changes on the epigenetic regulation of ruminant mammary gland function. As part of this line of research, Zhang et al. introduced an infusion of casein, arginine, and alanine into the diet of experimental cows. The results of their in vitro experiment showed differences between the experimental and control groups regarding the expression of nine miRNAs: miR-743a, miR-543, miR-101a, miR-760-3p, miR-1954, miR-712, miR-574-5p, miR-468, and miR-875-3p, while the in vivo research showed that arginine infusion contributed to increasing casein levels by enhancing the expression of the CSN1S1 and CSN1S2 genes. Based on the miRNA expression data, they suggested that, by reducing mitochondrial malate dehydrogenase activity, the upregulation of miR-743a in response to arginine inflow might enhance the casein yield [285,286]. miR-543 targets and inhibits SIRT1 and class III histone deacetylase [287]. Proline dehydrogenase (PRODH) and ornithine aminotransferase (OAT) have been found to be putative target genes for miR-760-3p and miR-1954, respectively. The higher expression of miR-574-5p and miR-712 in response to arginine intake suggests that their effects on mammary cells may be related to stimulating their proliferation, thereby further inducing casein synthesis in this group [285]. A similar study was previously conducted by Wang et al. (2016) by studying the differences in the effects of high- and low-quality diets on the miRNA-induced mechanisms of milk-protein production regulation in dairy cows. They found that low-cost feed rations (corn straw and rice straw) increased the expression of miRNAs such as let-7e, miR-375, and miR-17-3p and decreased that of miR-148b, miR-183, miR-21-3p, miR-874, miR-99a-5p, and let-7c [288].

Mobuchon et al. (2017), via RT-qPCR-based expression profiling, identified two miRNAs in the mammary gland whose expression was suppressed when sunflower oil was added to the diet of cows, namely miR-20a-5p and miR-142-5p, and to establish their functional roles, they identified their target genes. Among these, 23 predicted targets for miR-20a-5p and miR-142-5p were identified using the DIANA software. It is known that miR-20a-5p can act on lipoprotein metabolism by directly modulating APOBEC4 (apolipoprotein B mRNA-editing enzyme catalytic polypeptide-like 4), LDLR (low-density lipoprotein receptor), and VLDLR (very low-density lipoprotein receptor), which participate in fatty acid desaturation (by targeting SCD5 (stearoyl-CoA desaturase 5)) and lipid secretion (by acting on BTN1A1 (butyrophilin subfamily 1 member A1), the main component of the membranes of milk-fat globules). LPIN1 (lipin 1), which is strongly expressed in the mammary glands of cows during lactation and is implicated in the buildup of triacylglycerol in adipose tissue, is also a possible target of miR-20a-5p [289,290]. miR-142-5p, in turn, targets two isoforms of the ACSL enzyme (long-chain acyl-CoA synthetase). ACSL1 has been shown to be the main isoform in the mammary glands of lactating cattle [283], and it activates newly synthesized fatty acids before they can be metabolized or injected into lipid droplets [291]. In addition, as noted above, miR-142-5p can also promote milk-fat metabolism by inhibiting CTNNB1 expression [136]. Interestingly, how both of these miRNAs potentially target ELOVL6, a member of the fatty acid elongase family involved in lipid metabolism, remains unknown [290,292]. Previously, Li et al. (2015) conducted a similar experiment to study the role of miRNAs in mammary lipogenesis, choosing linseed and safflower oils as dietary supplements. Both supplements increased the expression of miR-98, miR-148b, miR-199a-3p, miR-199c, miR-21-5p, and miR-378 and decreased the expression of miR-200a. At the same time, on day 7, compared to the control period, only one miRNA, miR-486, was significantly affected by the addition of safflower oil, while no miRNA was significantly affected by the addition of flaxseed oil [52]. As noted in the previous sections, the increased expression of miR-200a suppresses the mRNA expression of the genes involved in the formation of fat drops [95]. Targeted analysis has shown that the stearoyl-CoA desaturases SCD1 and SCD5, which are involved in LCD biosynthesis and target miR-200a and miR-199a-3p, respectively. FADS2, which causes LCD desaturation, is a direct target for miR-98a, whereas miR-378 can regulate adipocyte differentiation by directly affecting PPARγ and C/EBPα (CCAAT/enhancer-binding protein α), which promotes lipogenesis and increases the lipid-droplet size in developing adipocytes upon overexpression [52,293]. It is also interesting that six of the seven major miRNAs were activated, which could be expected to repress a large number of mRNAs associated with FA synthesis. This expectation is consistent with another study in which it was found that feeding conjugated linoleic acid (CLA) suppressed milk-fat synthesis, which was accompanied by the inhibition of the expression of many genes involved in milk-lipid synthesis [294]. The SCD1 gene, presumably targeting miR-199a-3p, is a key gene that plays a role in USFA synthesis, reported to be suppressed in response to linseed oil supplementation [295].

Thus, deciphering the functions of miRNAs and, in particular, their effects on nutrition-regulated genes in the mammary gland can be of great importance for understanding the effects of diet on mammary gland development and, consequently, the regulation of milk synthesis and secretion.

## 5. Conclusions

Herein, our comprehensive review of the most innovative research work suggests that miRNAs play an important role in many processes related to breast development and health and disease, as well as in the processes of milk secretion and lactation. As we have learned, the mammary gland is a complex organ that is responsible for an equally complex lactation process regulated by a multitude of epigenetic elements, no small part of which are miRNAs. In the first chapter, we were able to understand that all of this explains the presence of multiple miRNAs in mammary gland tissues, the identification base for which is constantly replenished. However, as we considered further, we also realized that, even compared to the number of identified mammary gland miRNAs, there are very few known miRNAs whose functions are thoroughly understood. Therefore, the proportion of what we know about the regulatory networks underlying mammary gland function and the lactation process is very small relative to what may actually be occurring. The problem is that, since one miRNA can target hundreds of genes, the functional validation of each miRNA target gene requires expensive and time-consuming approaches. Therefore, to identify and study the potential role of miRNAs in mammary gland development and lactation biology, it would be rational to resort to integrated “omics” approaches (e.g., genomics, epigenomics, transcriptomics, and proteomics). Integrated approaches, such as a combination of miRNA and mRNA expression in a single sample, will improve computational predictions and, thus, our understanding of the function and application of miRNAs. These integrated “omics” approaches should also be used to identify and investigate the potential role of miRNAs in mammary gland development and lactation biology. Thus, the prospect of using miRNAs to improve mammary gland health and milk productivity, as well as milk quality, judging by the increasing number of published reports in the last few years, is becoming very promising.

## Figures and Tables

**Table 1 ncrna-07-00078-t001:** A brief summary of miRNAs ever found in bovine mammary glands.

Tissue Location	Amount of Found miRNAs	Identification Approach	References
Adipose and epithelial mammary gland tissues	59	Small RNA cloning	[47]
Raw milk and colostrum	441	Solexa Sequencing + MIREAP	[33]
Colostrum and mature milk	153	Microarray Analysis	[49]
Mammary gland biopsy	884	Solexa Sequencing	[50]
Mammary gland biopsy	167	Illumina Sequencing	[51]
Mammary epithelial cell line	344	RNA-Seq + miRDeep2	[51]
Mammary gland biopsy	497	Illumina Sequencing + miRDeep2	[52]
Milk fat	243	Illumina HiSeq 2000	[53]
Milk whey	231	Illumina HiSeq 2000	[53]
Milk cells	285	Illumina HiSeq 2000	[53]
Mammary gland tissue	497	Illumina HiSeq 2000	[53]
Mammary gland tissue	54	miRNA microarray analysis	[54]
Mammary gland cell cultures	408	Solexa Sequencing and bioinformatic Analysis of Small RNAs	[55]
Milk fat	713	Illumina HiSeq 2000	[56]
Mammary gland tissue	957	Illumina-Solexa high-throughput sequencing	[57]
Mammary gland tissue	326	Solexa Sequencing	[58]
Mammary epithelial cell line	344	RNA-Seq + miRDeep2	[59]
Mammary gland tissue	1838	RNA-Seq + miRDeep2	[60]
Milk exosomes	1472	Illumina Hiseq 2500	[61]
Udder biopsy	48	HiSeq2500 + Bowtie + miRDeep2	[62]
The mammary gland epithelium	546	Illumina Sequencing + Bioinformatic Analysis (DESeq2 R package)	[63]

**Table 2 ncrna-07-00078-t002:** A brief summary of all miRNAs ever found in goat mammary glands.

Tissue Location	Amount of Found miRNAs	Identification Approach	References
Mammary gland tissues	431	Illumina/Solexa high-throughput sequencing	[38]
Mammary gland tissues	441	Illumina/Solexa high-throughput sequencing	[67]
Colostrum	131	Solexa sequencing + bioinformatic analysis	[68]
Mammary gland tissues	1144	Solexa sequencing + bioinformatic analysis	[69]
Mammary gland tissues	4038	Illumina/Solexa high-throughput sequencing	[70]
The secretory area containing lobulo-alveolar structures (acini)	924	Illumina HiSeq 2500 + miRDeep2 + comparing with goat sequences	[71]
The secretory area containing lobulo-alveolar structures (acini)	1178	Illumina HiSeq 2500 + miRDeep2 + comparing with cattle sequences	[71]
Mammary gland (in silico)	29	Expressed sequence tag analysis and genome sequence analysis	[72]
Mammary gland tissues	1487	Illumina Sequencing + mapping of the mammalian miRNAs precursor sequences	[73]

**Table 3 ncrna-07-00078-t003:** A brief summary of all miRNAs ever found in ovine mammary glands.

Tissue Location	Amount of Found miRNAs	Identification Approach	References
Mammary epithelial tissue	50	Comparison with other species miRNomes	[76]
Parenchyma of the mammary gland	147	Illumina HiSeq 2500 + aligning with known miRNA sequences	[39]

**Table 4 ncrna-07-00078-t004:** A brief summary of the temporal patterns of miRNAs expression depending on the period of mammary gland development.

Animal Species	Period	Amount of miRNAs	Most Expressed miRNAs	Expression Pattern	References
Cattle	Peak lactation	165		Suppressed	[50]
	Lactation	9		Expressed	
	Dry period	6		Expressed	
	Dry period	12	miR-10a, miR-15b, miR-16, miR-21, miR-33b, miR-145, miR-146b, miR-155, miR-181a, miR-205, miR-221, miR-223	Suppressed	[45,67]
	Beginning of lactation	1	miR-31	Expressed	
	Lactation	15	miR-30a-5p, miR-30d, miR-21-5p, miR-26a, miR-148a, let-7a-5p, let-7b, let-7f, let-7g, miR-99a-5p, miR-191, miR-200a, miR-200c, miR-186, miR-92a	Expressed	[79]
	Post-pubertal	54	miR-10b, miR-29b, miR-101, miR-375, miR-2285t, miR-146b, let7b, miR-107, miR-1434-3p	Expressed	[54]
	Early lactation	1	miR-221	Expressed	[80]
	Dry period	12	miR-10a, miR-15b, miR-16, miR-21, miR-33b, miR-145, miR-146b, miR-155, miR-181a, miR-205, miR-221, miR-223	Suppressed	
	Pregnancy	1	miR-146b	Expressed	[67]
Sheep	Early, intermediate, late pregnancy; lactation	2	miR-21, miR-25	Suppressed	[86]
	First half of pregnancy	6	miR-205, miR-200a, miR-200b, miR-200c, miR-141, miR- 429	Expressed	
	Later pregnancy	1	miR-205	Suppressed	
	Non-lactating, peak-lactation	136		Expressed	[76]
	Non-lactating		miR-143, miR-21, miR-26a, miR-99a, miR-148a, let-7i, let-7g, let-7f, miR-199a-3p, miR-221, miR-125b, miR-329b-3p, miR-493-5p	Expressed	
	Peak-lactation		miR-148a, miR-143, miR-26a, let-7f, let-7g, miR-30a-5p, let-7a, miR-21	Expressed	
Goat	Peak lactation	8	miR-451, miR-2478, miR-2887	Expressed	[39]
	Dry period	12	miR-25, miR-128, miR-93, miR-98, miR-145, miR-199b, miR-199a-3p, miR-181b, miR-222, miR-221, let-7b, let-7c	Expressed	
	Peak lactation	165		Suppressed	
	Pregnancy, lactation, dry period	2988	miR-148a-3p, miR-30d	Expressed	[67]
	Dry period	221		Expressed	
		260		Suppressed	
	Late lactation	185		Expressed	
		247		Suppressed	
	Early lactation and dry periods		miR-148a, miR-17-5p	Expressed	[70]
	Colostrum lactation	45	miR-223-3p, miR-223-5p	Expressed	
	Colostrum lactation	86		Suppressed	
	Peak lactation	31	miR-30e-5p, miR-15a	Expressed	
		24		Suppressed	
	Lactation		miR-15b	Suppressed	
	Lactation		miR-135b	Expressed	[81]
	Lactation		miR-27a	Expressed	
	First half of pregnancy		miR-205	Expressed	[68]
			miRNA 200	Suppressed	
	End of pregnancy			Expressed	

**Table 5 ncrna-07-00078-t005:** A brief summary of the reciprocal regulation of miRNAs and various hormones.

Direction	miRNAs	Hormones	Influence	References
Hormone—miRNA	miR-23a, miR-27a, miR-27b, miR-103, miR-200a	Prolactin	Boost	[76,85,95]
	miR-183	Prolactin	Inhibit	[96]
miRNA—hormone	miR-135b	Prolactin	Inhibit	[84]
	miR-138	Prolactin	Inhibit	[97]
	miR-135a	Prolactin	Inhibit	
Hormone—miRNA	miR-21-5p, miR-25, miR-26a, miR-223, miR-320a, miR-339a, miR-148a	Dexamethasone, bovine insulin, sheep prolactin	Inhibit	[94]
Hormone—miRNA	miR-15b	Estradiol, progesterone	Inhibit	[83]
miRNA—hormone	miR-126-3p	Progesterone	Inhibit	[98]
miRNA—hormone	miRNA-15a	Growth hormone	Inhibit	[101]

**Table 6 ncrna-07-00078-t006:** A brief summary of the involvement of several miRNAs in lipid metabolism.

miRNAs	Regulated Genes/Proteins/Pathways	Regulation Matter	References
miR-103	AMPKα pathway	The ratio of unsaturated/saturated fatty acids in milk	[110]
miR-128-1, miR-148a, miR-130b, miR-301b	LDL, LDLR, ABCA1 cholesterol transporter	Cholesterol-lipoprotein trafficking	[111]
miR-24	FASN, SREBF1, SCD, GPAM, ACACA	Triacylglycerol content, unsaturated fatty acid concentration	[113]
miR-200a	ADRP, TIP47	Fat droplet formation	[95]
	SLC27A6, CD36	Fatty acid uptake	
	ACACA, FASN	Fatty acid synthesis	
	SCD, DGAT1	Triglyceride synthesis	
miR-212-5p, miR-27a, miR-27b, miR-132, miR-191, miR-214	SCD1, FAS	Triacylglycerol content	[114]
miR-135b	LATS2	Triacylglyceride and unsaturated fatty acids synthesis	[84]
miR-497			[115]
miR-16a	LATS1	TAG and cholesterol metabolism	[116]
miR-375	ERK1/2, PPARγ	Fat cell formation and differentiation	[117]
miR-146b	SIRT1	3T3-L1 cells adipogenesis	[86]
miR-204-5p, miR-141	SIRT1, SREBF1, FASN, PPARγ	Lipid synthesis	[119,120]
miR-212	SIRT2, FASN, SREBP1	Increasing the fat content in mammary epithelial cells	[121]
miR-106b	ABCA1	Milk fat synthesis	[122]
		The accumulation of triglycerides and cholesterol in epithelial cells of the mammary gland	[123]
miR-30e-5p, miR-15a	LRP6, YAP1	Promoting the fat metabolism, mediating adipocytes differentiation	[124]
miR-126-3p	FASN	Lipid synthesis in the mammary gland	[125]
miR-150		Lipogenesis inhibition in mammary epithelial cells	[126]
miR-145	INSIG1	Facilitates milk fat synthesis	[65]
miR-15b	FASN	Lipid metabolism suppression	[83]
miR-34b	FASN, FABP4, C/ EBPα, DCP1A	TAG accumulation inhibition and lipid droplet formation suppression	[127]
miR-33a	ELOVL5, ELOVL6, SC4MOL	Fatty acid oxidation	[128]
miR-152	PTGS2, PRKAA1, CUP3	Prostaglandin synthesis	
miR-224	LPL, GST, ALOX15, PTGS1	Milk fat metabolism	
miR-221	FASN, ACSL1, ElF5, NR1H3	Lipid droplet formation	[137]
miR-143	PPARγ, FASN, SCD1, CEBPβ, SREBP1	Lipid droplet formation	[138]
miR-183	MST1	Milk fat metabolism	[140]
	IRS1		[96]
miR-124a	PECR	Fatty acid metabolism	[55]
miR-193a-5p	FADS1	Milk fatty acid content	[142]

**Table 7 ncrna-07-00078-t007:** A brief summary of the participation of some miRNAs in the cellular processes of mammary tissues.

miRNAs	Regulated Genes/Proteins/Pathways	Regulation Matter	References
miR-101a	COX-2	Mammary gland differentiation and involution	[148]
miR-200a		Epithelial cell differentiation	[76,149,150]
miR-205		Cell size, cell proliferation	[151]
miR-24-3p	MEN1	Epithelial cell proliferation	[153]
miR-221	STAT5a, STAT3 JAK-STAT, IRS1	Cell proliferation	[80]
miR-143-3p	Ndfip1	Mammary epithelial cells apoptosis	[158]
miR-145	FSCN1	Mammary epithelial cell proliferation	[159]
miR-15a	YAP	Adipocyte growth and apoptosis	[124]
miR-25	Kruppel-like factor 4, CCAAT/binding enhancer protein alpha	Adipocyte differentiation	[167]

**Table 8 ncrna-07-00078-t008:** A brief summary of some miRNAs roles in the control of the mammary immune system.

miRNAs	Regulated Genes/Proteins/Pathways	Regulation Matter	References
miR-21, miR-146a, miR-155	Toll-like receptor, NF-kB		[175]
miR-30a-5p, miR-30d-5p	JAK/STAT pathway, IL-10		[176]
miR-22-3P		Development and differentiation of T-lymphocytes	[177]
miR-19b, miR-23b-3p, miR-331-5p, miR-664b, miR-2431-3p	GNG2, MAPRE2, CD14, IL17A, S100A9, COL4A1, RAP1B, LDOC1, LDLR	Inflammation and immunity	[186,187,188,189,190,191]
miR-15a, miR-16a, miR-21-3p, miR-29b, miR-125b, miR-181a, miR-148a, miR-223, miR-375, let-7f			[183,193]
miR-200a, miR-205, miR-122, miR-182 conservative_15_7229	Toll-like receptor signaling pathway Chemokines signaling pathway T-cell receptor signaling pathway		[197]
miR-21 family	CALB, Vitamin D-dependent antimicrobial pathway	Inflammatory response to mastitis	[198,199,201,202,203,204]
miR-223	CBLB	Mammary alpha T cells stimulation	[62]
miR-145	FSCN1	Levels of cellular immune cytokines	[208]
miR-15a	IRAK2	Inflammation and differentiation of immune cells	[206,207,208]
miR26a	FGA	Maintenance of immune and defense responses, cell proliferation and apoptosis, and tissue injury and healing	[58]
let-7d, let-7b, mir-98, miR-100, mir-130a, miR-193a, miR-210, miR-494, miR-652	MAPK, JAK-STAT	*S. uberis* polysaccharide-induced immune response	[210]
miR-155, miR-146a, miR-222, miR-383, miR-21	NF-KB and MAPK Signaling Pathways	Mastitis-induced inflammation	[240]
miR-125b	NFκB (TRAF6 and A20)	Inflammatory response	[203]
miR-223	IKKα		
miR-29a/miR-29b	IFN-γ	Mastitis-induced inflammation	[210]
miR-101, miR-142-5p, miR-183, miR-2285g-3p, miR-223, miR-99a-5p	22 genes	Immune responses with *S. aureus* infection	[184]

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
