# Peer review of "The Role of microRNAs in the Mammary Gland Development, Health, and Function of Cattle, Goats, and Sheep"

_ncrna, 2021, doi:10.3390/ncrna7040078_

Round 1

Reviewer 1 Report

The work is a very extensive collection of scientific articles recently
published on the role of microRNAs in farm animals at the level of mammary
gland development, and health. The main issue of this review is that,
although the work is very thorough at the bibliography level, it does
not go into details on the role of miRNAs as RNAs that regulate their own
target genes by affecting their molecular pathways. I suggest to include some examples of
examples of miRNAs that deregulate cellular molecular pathways
leading to chronic or acute inflammation for example.
A minimal revision of English
is required. I also suggest removing those sentences that were
inadvertently copied and pasted from the web page template while
submitting the article. 

I therefore recommend these revisions before considering this draft for publication

P.S Plagiarism software detected 11% of plagiarism

Author Response

Point 1:  I suggest to include some examples of
examples of miRNAs that deregulate cellular molecular pathways
leading to chronic or acute inflammation for example.

Response 1: We accepted this recommendation while also satisfying a similar one by reviewers 2 and 3, and decided to compile tables following some of chapters, including the section on miRNAs role in the immune response. We attempted to reflect the necessary data in this table.

Point 2: A minimal revision of English is required.

Response 2: Done. To achieve this, we used the MDPI English Editing service. You can identify changes in the manuscript text by highlighting in the review mode (blue).

Point 3: I also suggest removing those sentences that were inadvertently copied and pasted from the web page template while submitting the article.

Response 3: Done. Thank you for your remark, and please accept my apologies for such blunder.

Reviewer 2 Report

The authors provided an extensive review on the role that miRNAs play in mammary gland development, health, and function of farm animals. The authors did a fantastic job on reviewing so many papers and on giving such a broad overview on several miRNAs linked to the mammary gland of cattles, sheeps, and goats. There are a few minor points that need to be addressed before the manuscript is considered suitable for publication.

Please change the title and list only the three species that are described in this review. The term “farm animals” comprises also other animals, such as pigs, chickens, and other species, that are not described in this review.  

At the beginning of a few paragraphs there is a section with author guidelines that needs to be deleted (for example, Introduction, lines 25-33).

Sometimes, the use of English is hard to follow, and I would recommend to review the entire manuscript (perhaps with the help of a native English speaker, if feasible) and rephrase a few sentences (for example: line 170-171, line 209-217, line 915-916).

Throughout the manuscript there are inconsistencies on how miRNAs are called. The authors refer to them using different nomenclatures: “microRNA-#”, “miR-#”, “miRNA-#”, “just the number”, “bta-miR-#”. Would it be possible to be consistent and choose only one nomenclature?

Line 219-220: please change the term “object” with a term that is scientifically more suitable, when referring to the mammary gland (perhaps “tissue” / “organ”?).

Line 252: what are “readings”?

Line 758, 761, 762: please change “NF-jB” with “NF-kB”.

Line 836-839: it looks like there is a section of that sentence that is repeated twice.

Author Response

Point 1: Please change the title and list only the three species that are described in this review. The term “farm animals” comprises also other animals, such as pigs, chickens, and other species, that are not described in this review.

Response 1: Done. We have just listed the three species mentioned in the manuscript.

Point 2: At the beginning of a few paragraphs there is a section with author guidelines that needs to be deleted (for example, Introduction, lines 25-33).

Response 2: Done. Thank you for your remark, and please accept my apologies for such blunder.

Point 3: Sometimes, the use of English is hard to follow, and I would recommend to review the entire manuscript (perhaps with the help of a native English speaker, if feasible) and rephrase a few sentences (for example: line 170-171, line 209-217, line 915-916).

Response 3: Done. To achieve this, we used the MDPI English Editing service. You can identify changes in the manuscript text by highlighting in the review mode (blue).

Point 4: Throughout the manuscript there are inconsistencies on how miRNAs are called. The authors refer to them using different nomenclatures: “microRNA-#”, “miR-#”, “miRNA-#”, “just the number”, “bta-miR-#”. Would it be possible to be consistent and choose only one nomenclature?

Response 4: Done. However, it should be emphasised that miRNA's nomenclature in our manuscript was determined by how a miRNA was referenced in the original article. As a result, we didn't alter its name at first time.

Point 5: Line 219-220: please change the term “object” with a term that is scientifically more suitable, when referring to the mammary gland (perhaps “tissue” / “organ”?).

Response 5: Done. Changed with an "organ". Thank for your remark.

Point 6: Line 252: what are “readings”? 

Response 6: Changed with an "reads". Read is an inferred sequence of base pairs corresponding to all or part of a single DNA fragment. However, for us, there was a translation error resolved thanks to you.

Point 7: Line 758, 761, 762: please change “NF-jB” with “NF-kB”.

Response 7: Done.

Point 8: Line 836-839: it looks like there is a section of that sentence that is repeated twice.

Response 8: Fixed.

Reviewer 3 Report

The manuscript by Artem P. Dysin et al. this study evaluated the role of microRNAs in mammary gland development, health, and function of farm animals. Milk is an integral and therefore a complex structural element of mammalian nutrition. Therefore, it is simple to conclude that lactation, the process of its producing, is as complex as the mammary gland, the organ in charge of this biochemical activity. The significance of miRNAs in signaling pathways, cellular proliferation and lipid metabolism in agricultural ruminants, which are crucial in light of their role in the nutrition of human as a consumer of dairy products, will be discussed. It is a topic of interest to the researchers in the related areas. The article is acceptable after some revision.

  1. Some paragraphs are lengthy,such as line123-171, line699-799, etc, it is recommended to reflect the key points in paragraphs.
  2. Some words that have already appeared abbreviated don't need to be repeated. For example, line63 has already mentioned non-coding RNA (ncRNA), and line66 has appeared non-coding RNA. The same problem also appears in line96 "microRNA".
  3. The word "miRNA" is not suitable to appear at the beginning of a sentence alone, such as line 126.
  4. The author has described each chapter in as much detail as possible, but too much data will lead to the lack of key points. You should consider replacing part of the data with charts. And there should be some summary and discussion after each chapter
  5. In the conclusion, the author focused more on the research method, and rarely mentioned the "role of miRNA" in the title. Similarly, the outlook for the future is rarely mentioned.

Author Response

Point 1: Some paragraphs are lengthy,such as line123-171, line699-799, etc, it is recommended to reflect the key points in paragraphs.

Response 1: We actually found some of our paragraphs too long and difficult to read, so we split them into smaller sections. In addition, in several of the chapters, we built a tables that highlighted essential information on some of the considered miRNAs. However, the overall size of the chapters and the volume of the text of our manuscript is due to the large number of research papers that we are considering, as well as the necessity for an in-depth analysis of the major subject of each chapter.

If you consider some parts should be compacted, please, can you hint us where we should focus our attention?

Point 2: Some words that have already appeared abbreviated don't need to be repeated. For example, line63 has already mentioned non-coding RNA (ncRNA), and line66 has appeared non-coding RNA. The same problem also appears in line96 "microRNA".

Response 2: Fixed.

Point 3: The word "miRNA" is not suitable to appear at the beginning of a sentence alone, such as line 126.

Response 3: Please could you explain why? This was not identified as a mistake by the MDPI English Editing service.

Point 4:The author has described each chapter in as much detail as possible, but too much data will lead to the lack of key points. You should consider replacing part of the data with charts. And there should be some summary and discussion after each chapter

Response 4: As noted in the previous response, we followed your recommendation and indeed represented some of the data in tables to make it simpler to comprehend.However, we performed it via addition rather than replacement, because some readers find it more comfortable to refer to the text. However, if you do not agree with this decision, we will consider ways to make the text more concise.

As for the summary and discussion, we have divided them into distinct paragraphs and slightly enhanced them.

Point 5: In the conclusion, the author focused more on the research method, and rarely mentioned the "role of miRNA" in the title. Similarly, the outlook for the future is rarely mentioned.

Response 5: The manuscript is entirely devoted to the role of miRNA in biological processes occurring in the mammary gland, therefore the "role" is first of all mentioned in the title of the work. If you believe this should be mentioned more frequently, please suggest where in the manuscript this should be done. In terms of focusing on research methodology, we considered this aspect to be inherent since miRNA is a relatively new discovery in molecular biology and, as a result, necessitates new technical approaches to study. Please also explain which sections of the paper would be acceptable for mentioning the future outlook. Thank you for your report as well.